# DNA methylation signatures of monozygotic twins clinically discordant for multiple sclerosis

Nicole Y. Souren[1], Lisa A. Gerdes[2], Pavlo Lutsik [3], Gilles Gasparoni[1], Eduardo Beltrán[2], Abdulrahman Salhab[1], Tania Kümpfel[2], Dieter Weichenhan[3], Christoph Plass [3], Reinhard Hohlfeld[2,4] & Jörn Walter[1]

Multiple sclerosis (MS) is an inflammatory, demyelinating disease of the central nervous system with a modest concordance rate in monozygotic twins, which strongly argues for involvement of epigenetic factors. We observe highly similar peripheral blood mononuclear cell-based methylomes in 45 MS-discordant monozygotic twins. Nevertheless, we identify seven MS-associated differentially methylated positions (DMPs) of which we validate two, including a region in the *TMEM232* promoter and *ZBTB16* enhancer. In CD4 + T cells we find an MS-associated differentially methylated region in *FIRRE*. Additionally, 45 regions show large methylation differences in individual pairs, but they do not clearly associate with MS. Furthermore, we present epigenetic biomarkers for current interferon-beta treatment, and extensive validation shows that the *ZBTB16* DMP is a signature for prior glucocorticoid treatment. Taken together, this study represents an important reference for epigenomic MS studies, identifies new candidate epigenetic markers, and highlights treatment effects and genetic background as major confounders.

[1] Department of Genetics/Epigenetics, Saarland University, 66123 Saarbrücken, Germany. [2] Institute of Clinical Neuroimmunology, University Hospital and Biomedical Center, Ludwig-Maximilians University Munich, 81377 Munich, Germany. [3] Division of Cancer Epigenomics, German Cancer Research Center (DKFZ), 69120 Heidelberg, Germany. [4] Munich Cluster for Systems Neurology (SyNergy), 80336 Munich, Germany. Correspondence and requests for materials should be addressed to N.Y.S. (email: nicole.souren@hotmail.com) or to J.W. (email: j.walter@mx.uni-saarland.de)

Multiple sclerosis (MS), a leading cause of neurological disability in young adults, is considered to be an autoimmune disease, characterized by chronic inflammatory demyelination of the central nervous system[1,2]. Although nuclear genetic factors contribute to the development of MS[3], a maximum concordance rate for MS in monozygotic (MZ) twins of 25%[4,5], indicates that interaction with other risk factors is compulsory for clinical symptoms to develop. While various studies suggested mitochondrial DNA variants as plausible MS susceptibility factors, we recently showed that mitochondrial DNA variation (e.g., skewed heteroplasmy) does not play a major role in the discordant clinical manifestation of MS in MZ twins[6].

DNA methylation differences represent another source of molecular variation that can cause discordant phenotypes within MZ twins[7–13]. As DNA methylation changes can cause transcriptional alterations, aberrant DNA methylation has been observed in various human diseases[14,15]. Discordant DNA methylation profiles within MZ twins have been reported quite frequently at imprinted regions[7–9], which are characterized by parent-of-origin-specific methylation patterns resulting in monoallelic expression. As a maternal parent-of-origin effect in MS susceptibility has been reported[16,17], and several imprinted genes have been linked to immune system development and functioning (reviewed by Ruhrmann et al.[18]), genomic imprinting errors might be involved in the pathogenesis of MS[18]. Additionally, environmental risk factors such as smoking, history of symptomatic Epstein-Barr virus infection, and vitamin D deficiency have been associated with an increased MS risk[19–21]. Although the molecular mechanisms underlying these associations remain unknown, evidence that these environmental factors can induce DNA methylation changes is accumulating[22–25].

Thus far, several epigenome-wide association studies (EWAS) for MS have been carried out[26–31], and a number of differentially methylated CpG positions (DMPs) have been reported, including DMPs in the *HLA-DRB1* locus. Although these studies used the same array platform (i.e., Infinium HumanMethylation450 (450 K)), the results are inconsistent. Since these studies used genetically unmatched cases and controls, they are potentially hampered by DNA sequence variation. As genetic factors predispose to MS, these studies cannot determine whether MS is due to genetic or epigenetic susceptibility. In addition, SNP-containing probes give rise to biased DNA methylation measurements[32], and DNA methylation changes are also often the result of *cis*- or *trans*-acting genetic variants (methylation quantitative trait loci or mQTLs)[33]. A MZ twin-based design controls for these genetic differences and for other factors (potentially) affecting the methylome, including gender, age, and a broad range of environmental factors. Thus far, one EWAS in MS-discordant MZ twins has been reported, but no DNA methylation differences were identified[34]. Since this study included only three pairs and exclusively aimed at identifying very large methylation differences (i.e., ≥80% methylation in one co-twin and ≤20% in the other), further studies in larger cohorts are required.

Here, we describe an EWAS comprising a unique cohort of 45 MZ twins clinically discordant for MS in which we aim to identify MS-associated DNA methylation changes in peripheral blood mononuclear cells (PBMCs) and to study the effect of MS treatments on the methylome (Fig. 1). Although we confirm that MS-discordant MZ twins have very similar methylomes, we identify a few new MS-associated candidate loci and observe DNA methylation changes associated with current interferon-beta (IFN) and prior glucocorticoid (GC) treatment.

## Results

**PBMC-based methylomes.** PBMCs of 46 MZ twins clinically discordant for MS were accessible and genome-wide DNA methylation profiles were established using Illumina Infinium MethylationEPIC BeadChips (EPIC arrays). After quality control and filtering, methylation data of 849,832 sites were available for 45 twin pairs. As expected, within-pair array-wide correlation coefficients were very high (mean = 0.995), indicating high-quality data. Clinical characteristics of the 45 MS-discordant MZ twins are shown in Table 1 and Supplementary Fig. 1.

**Detection and validation of MS-DMPs in *TMEM232* and *ZBTB16*.** To identify DMPs associated with the clinical manifestation of MS (MS-DMPs), first a pair-wise analysis was carried out on the EPIC array data of the 45 pairs without adjusting for cell-type composition (Supplementary Fig. 2a, b). The Q-Q plot in Supplementary Fig. 2b shows that the obtained *p*-values (Wilcoxon singed-rank test) clearly deviate from the null expectation. This inflation was eliminated after adjusting for cell-type composition (Fig. 2a, b), indicating that many differences are due to variation in cellular composition.

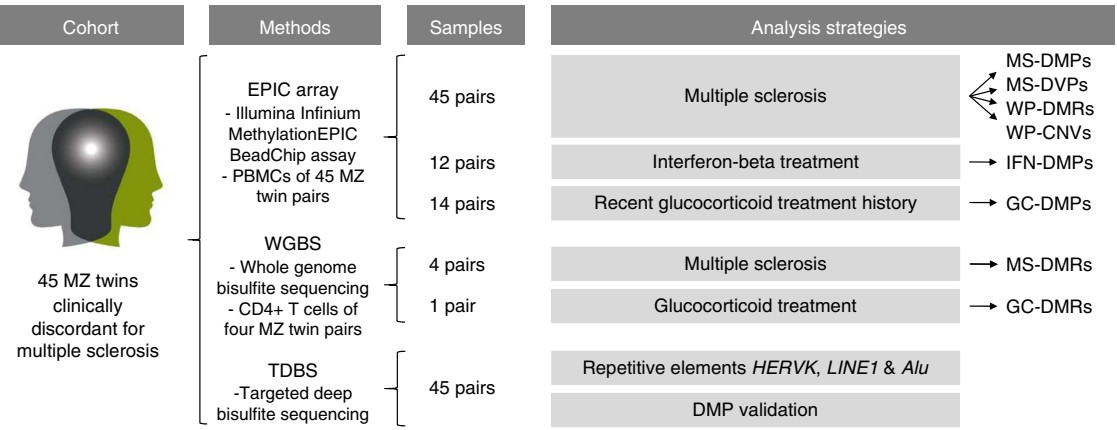

**Fig. 1** Schematic overview of the study design and analysis strategies. DMPs, differentially methylated CpG positions; DMRs, differentially methylated regions; DVP, differentially variable CpG positions; GC-DMPs, glucocorticoid treatment-associated DMPs; HERVK, human endogenous retrovirus type K; IFN-DMPs, interferon-beta treatment-associated DMPs; LINE1, long interspersed nuclear element-1; MS-DMPs, multiple sclerosis-associated DMPs; MS-DVPs, multiple sclerosis-associated DVPs; MZ, monozygotic; PBMCs, peripheral blood mononuclear cells; WP-CNVs, within-pair copy-number variations; WP-DMRs, within-pair differentially methylated regions. The logo of the MS/TWIN/STUDY is not covered by the article CC BY license. Image credit goes to Lisa Ann Gerdes. All rights reserved, used with permission

**Table 1 Characteristics of the MZ twins clinically discordant for MS**

| Characteristic | MS-affected MZ co-twins | Non-affected MZ co-twins | Range | $p^{b}$ |
|---|---|---|---|---|
| Number of pairs | 45 | 45 | | |
| Gender (female/male) | 32/13 | 32/13 | | |
| Age at study entry (years) | $42.3 \pm 12.1$ | $42.3 \pm 12.1$ | (21–67) | |
| Age of disease onset (years)[a] | $27.9 \pm 8.4$ | | (14–46) | |
| Years clinically discordant for MS at sample collection[a] | $15.3 \pm 11.1$ | | (1–45) | |
| EDSS at study entry | $3.3 \pm 2.3$ | | (0–9.5) | |
| Pairs longer than 10 years clinically discordant for MS | 25 (56%) | | | |
| Pairs with a positive family history of MS | 13 (29%) | | | |
| MS type | | | | |
| - RRMS | 31 (69%) | | | |
| - SPMS | 12 (27%) | | | |
| - PPMS | 2 (4%) | | | |
| Smoking status | | | | |
| Smoking at disease onset | 23 (51%) | 19 (42%) | | 0.53 |
| Pack-years at disease onset | 0.03 (0–3.5) | 0 (0–3.8) | | 0.81 |
| Smoking at sample collection | 14 (31%) | 12 (27%) | | 0.82 |
| Pack-years at sample collection | 0.6 (0–10.8) | 0 (0–6.3) | | 0.24 |

Continuous data expressed as: mean ± standard deviation or median (interquartile range). Categorical data expressed as: number of observations (%)
*EDSS* Expanded Disability Status Scale, *PPMS* primary-progressive MS, *RRMS* relapsing-remitting MS, *SPMS* secondary-progressive MS
[a]See Supplementary Fig. 1 for boxplots (with all data points) showing the distribution of the age of disease onset and the years that the MZ twins were clinically discordant for MS at sample collection
[b]MS-affected versus non-affected MZ co-twins calculated using a two-tailed Wilcoxon signed-rank test for continuous data and two-tailed Fisher's exact test for categorical data

Mean within-pair β-value differences (Δβ-values) were small (Fig. 2a and Supplementary Fig. 2a). The largest differences were observed for *ECT2* (cg12393503), *SELPG* (cg02520593), and *IL34* (cg01447350), with mean Δβ-values of 0.15, 0.06, and −0.09, respectively, but they did not reach statistical significance. In several twins, these CpGs showed very large Δβ-values (~0.8) (Supplementary Fig. 3), but these differences were not confirmed by validation using targeted, deep bisulfite sequencing (TDBS) (Supplementary Fig. 4). This indicates that some EPIC probes are prone to technical artefacts, as reported by others[35], and that validation using independent assays is required.

The unadjusted analysis revealed 39 MS-DMPs with a suggestive $p < 5 \times 10^{-6}$ (Wilcoxon singed-rank test). After correcting for multiple testing six MS-DMPs remained genome-wide significant (false discovery rate (FDR) < 0.05) (Supplementary Fig. 2a). After adjusting for cell-type composition, no MS-DMP had FDR < 0.05, but five MS-DMPs had a suggestive $p < 5 \times 10^{-6}$ (Fig. 2a and Table 2). One of these MS-DMPs is located in the promoter of the *TMEM232* gene (cg27037608, mean Δβ-value = 0.024), encoding for a transmembrane protein of unknown function. Genetic variants in *TMEM232* have been associated with atopic dermatitis and allergic rhinitis in GWAS[36,37]. For this MS-DMP, EPIC array data for 12 neighboring CpGs were also available, which all showed a similar effect, and $p < 0.01$ was calculated for eight CpGs (Fig. 2c). A second solitary MS-DMP was observed in the gene body of *SEMA3C* (cg00232450, mean Δβ-value = 0.013), which has been suggested to promote dendritic cell migration during innate and adaptive immune responses, and to be involved in axonal guidance and growth[38]. A third MS-DMP is located in the *YWHAG* gene (cg01708711, mean Δβ-value = 0.015), which has been associated with MS severity in a GWAS[39]. This MS-DMP has five neighboring CpGs on the array, but all are non-significant, questioning the significance of this MS-DMP. A fourth MS-DMP (cg25345365) showed the largest methylation difference (mean Δβ-value = −0.039) and is located in an enhancer within *ZBTB16*, which has been reported to be essential for natural killer T (NKT) cell development[40]. The fifth MS-DMP (cg25755428, mean Δβ-value = 0.033) is located in the *MRI1* gene, in which mutations have been associated with vanishing white matter disease[41]. The β-value distribution of this and neighboring CpGs suggests that this concerns a mQTL (Table 2). Additional adjustment for smoking status did not alter the $p$-values of these 5 MS-DMPs, indicating that they are not confounded by smoking status.

Next, all 698 MS-DMPs with $p < 0.001$ (Wilcoxon singed-rank test) after adjusting for cell-type composition were functionally annotated using the GREAT tool, which assigns biological meaning to a set of non-coding genomic regions by analyzing the annotations of nearby genes[42]. This analysis revealed that *TMEM232* is enriched for MS-DMPs. Other annotation categories were not significant.

Based on their significance, effect size and/or whether neighboring probes were also differentially methylated, the *TMEM232* (cg27037608) and *ZBTB16* (cg25345365) MS-DMPs were selected for validation using TDBS. The TDBS data correlated highly with the array data ($r_{\text{Pearson-}TMEM232} = 0.84$ and $r_{\text{Pearson-}ZBTB16} = 0.89$, Supplementary Figs. 5a and 6a), and both MS-DMPs as well as the surrounding CpGs were significantly differentially methylated between the MS-discordant twins (Supplementary Figs. 5b and 6b). This confirms that these MS-DMPs represent true effects in our cohort.

**TMEM232 and ZBTB16 MS-DMPs associated with long-standing MS.** To verify whether the identified methylation differences are dependent on disease duration, we performed a pair-wise analysis including only the EPIC array data of the 25 pairs that have been clinically discordant for MS longer than 10 years (Supplementary Table 1). In the analysis adjusted for cell-type composition, two DMPs had a suggestive $p < 5 \times 10^{-6}$ (Wilcoxon singed-rank test), one of which is located in the promoter of the *TACSTD2* gene encoding tumor-associated calcium signal transducer 2 (mean Δβ-value = −0.022)[43]. The second DMP is located in the promotor of the *RCL1* gene (mean Δβ-value = −0.011), which has been linked to depression[44]. For these two long-standing MS-DMPs, EPIC array data of neighboring CpGs were available ($< 500$ bp), and two neighboring CpGs of the *TACSTD2* MS-DMP showed a similar trend ($p < 0.10$). Additionally, the *TMEM232* MS-DMPs cg27037608 (mean Δβ-value = 0.026, $p = 3.2 \times 10^{-5}$) and cg26583412 (mean Δβ-value = 0.038, $p = 1.8 \times 10^{-5}$) were among the top 15 most significant DMPs associated with long-standing MS.

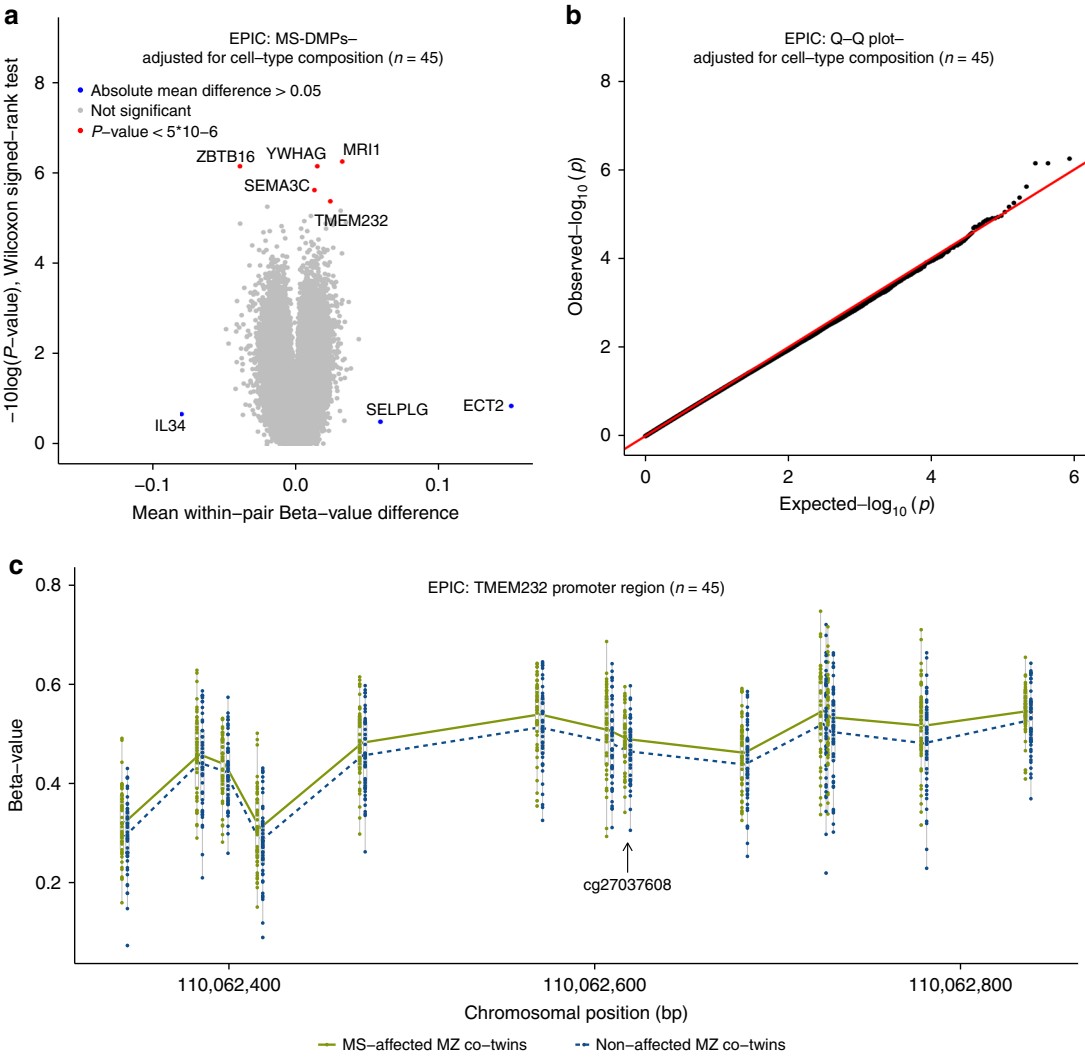

**Fig. 2** DNA methylation changes associated with the clinical manifestation of MS. Results of the differential DNA methylation analysis including the EPIC array data of the 45 MZ twin pairs clinically discordant for MS. **a** Volcano plot of the *p*-values resulting from the nonparametric two-tailed Wilcoxon signed-rank test against the mean within-pair β-value difference for each CpG. Data were adjusted for cell-type composition. **b** Q-Q plot of the *p*-values resulting from the nonparametric two-tailed Wilcoxon signed-rank test shown in Fig. 2a. Data were adjusted for cell-type composition. Within-pair β-value difference (Δβ-value) = clinically MS-affected MZ co-twin—non-affected MZ co-twin. **c** Overview of the *TMEM232* promoter region. Data are presented as Tukey boxplots including the individual data points that represent the (adjusted) β-values of the significant MS-associated differentially methylated CpG position (MS-DMP) cg27037608 and 12 neighboring CpGs present on the EPIC array. The lines connect the mean methylation values of each CpG site for the MS-affected and clinically non-affected MZ co-twins separately. Boxplots represent the interquartile range or IQR (bottom and top of the box) and 1.5 times the IQR (whiskers). Source data are provided as a Source Data file. *n* = number of twin pairs

Furthermore, the *ZBTB16* MS-DMP had a mean Δβ-value difference of −0.036 ($p = 0.002$, Supplementary Table 1). Additional adjustment for smoking status did not change the results. Hence, the *TMEM232* and *ZBTB16* MS-DMPs are also associated with long-standing MS.

**Evaluation of the *TMEM232* MS-DMPs in a case-control cohort.** Next, we evaluated the selected MS-DMPs in whole blood-based 450 K EWAS data of 140 unrelated MS patients and 139 controls from Kular et al.[31]. Unfortunately, the cg25345365 *ZBTB16*, cg27037608 and cg26583412 *TMEM232* MS-DMPs were not present on the 450 K array, but data from seven neighboring CpGs in *TMEM232* were available. Although none of these CpGs were significantly differentially methylated between the MS cases and controls ($p > 0.05$, linear regression), methylation levels were always higher in the MS patients, confirming the directionality of the effect observed in the twins (Supplementary Table 2).

**Whole-genome bisulfite sequencing reveals MS-DMR in *FIRRE*.** To identify additional MS-associated differentially methylated regions (MS-DMRs), we performed whole-genome bisulfite sequencing (WBGS) for a subset of four MS-discordant female twin pairs on CD4 + memory T cells, which have been implicated in the pathogenesis of MS[45]. First, genome-wide DNA methylation changes were evaluated by identifying and comparing partially methylated domains (PMD), fully methylated regions (FMRs), low methylated regions (LMRs), and unmethylated regions (UMRs). However, no significant differences were observed between the MS-discordant co-twins ($p > 0.05$, paired *t*-test) (Supplementary Figs. 7, 8).

Next, a DMR analysis was carried out and MS-DMRs were defined as ≥ 3 CpGs (max. distance 500 bp), each having $p < 0.05$ (paired *t*-test) and an absolute mean methylation difference >0.2. The DMR analysis revealed a prominent MS-DMR located in an intronic CTCF/YY1 bound regulatory region in *FIRRE*, which

**Table 2 DMPs associated with the clinical manifestation of MS (n = 45 twin pairs)[a]**

| Probe ID | Gene/ Location[b] | Functional region[c] | β-value MS | (U/A) non-MS | Δβ value (95% CI) (U/A) | β-value range | $p_{W-U}$/$p_{W-A}$ | $FDR_{W-U}$/ $FDR_{W-A}$ | Close probes[d] | 450 K | Full name & reported function |
|---|---|---|---|---|---|---|---|---|---|---|---|
| cg27037608 | *TMEM232*/ chr5: 110062618 | TSS200/ TFBS | 0.488/ 0.489 | 0.466/ 0.465 | 0.022 (0.012,0.032)/ 0.024 (0.014,0.034) | 0.31–0.59 | $4.8 \times 10^{-5}$/ $4.3 \times 10^{-6}$ | 0.13/ 0.74 | 13 within 500 bp, 8 with p < 0.01 | N | Transmembrane protein 232: associated with atopic dermatitis and allergic rhinitis in GWAS[36,37] |
| cg00232450 | *SEMA3C*/ chr7: 80421169 | Body/DHS | 0.813/ 0.810 | 0.793/ 0.797 | 0.020 (0.012,0.027)/ 0.013 (0.008,0.019) | 0.73–0.86 | $1.6 \times 10^{-7}$/ $2.5 \times 10^{-6}$ | 0.03/ 0.52 | 0 within 2 kb | N | Semaphorin 3 C: involved in axonal guidance and growth. Promotes dendritic cell migration during innate and adaptive immune responses[38] |
| cg01708711 | *YWHAG*/ chr7: 75959031 | Body/CpG island/ TFBS | 0.855/ 0.854 | 0.839/ 0.839 | 0.016 (0.011,0.021)/ 0.015 (0.010,0.020) | 0.79–0.89 | $1.8 \times 10^{-7}$/ $7.3 \times 10^{-7}$ | 0.03/ 0.21 | 5 within 350 bp, p > 0.01 | Y | Tyrosine 3-monooxyge-nase/ tryptophan 5-mono-oxygenase activation protein, gamma: associated with MS severity in GWAS[39] |
| cg25345365 | *ZBTB16*/ chr11: 114050114 | Body/ DHS/ FANTOM5 enhancer | 0.540/ 0.544 | 0.587/ 0.583 | −0.047 (−0.063, −0.031)/ −0.039 (−0.053, −0.024) | 0.36–0.72 | $1.5 \times 10^{-7}$/ $7.3 \times 10^{-7}$ | 0.03/ 0.21 | 0 within 2 kb | N | Zinc Finger And BTB Domain Containing 16: transcription factor essential for NKT cell development[40] |
| cg25755428 | *MRI1*/ chr19: 13875111 | TSS1500/ CpG island/ DHS | 0.328/ 0.336 | 0.311/ 0.303 | 0.017 (0.006,0.027)/ 0.033 (0.021,0.044) | 0.05–0.89[e] | $8.6 \times 10^{-4}$/ $5.8 \times 10^{-7}$ | 0.17/ 0.21 | mQTL[e] | Y | Methylthioribose-1-phosphate isomerase 1: includes mutation associated with vanishing white matter disease[41] |

Source data are provided as a Source Data file.
*A* adjusted for cell-type composition, *CI* confidence interval, *DHS* DNAse I hypersensitive site, $FDR_{W-A}$ FDR two-tailed Wilcoxon signed-rank test adjusted for cell-type composition, $FDR_{W-U}$ FDR two-tailed Wilcoxon signed-rank test unadjusted for cell-type composition, *GWAS* genome-wide association study, $p_{W-A}$ p-value two-tailed Wilcoxon signed-rank test adjusted for cell-type composition, $p_{W-U}$ p-value two-tailed Wilcoxon signed-rank test unadjusted for cell-type composition, *TFBS* transcription factor-binding site, *TSS200* the region from transcription start site (TSS) to −200 nt upstream of TSS, *TSS1500* −200 to −1500 nt upstream of TSS, *U* unadjusted for cell-type composition, *450 K* CpG present on 450 K array (*N* no, *Y* yes), *Δβ-value* within-pair β-value difference (clinically MS-affected MZ co-twin—non-affected MZ co-twin)
[a]Listed are the five MS-DMPs with a suggestive $p < 5 \times 10^{-6}$ (two-sided Wilcoxon signed-rank test) in the pair-wise analysis carried out using the EPIC array data of the 45 MZ twin pairs adjusted for cell-type composition
[b]Genome coordinates are human genome build GRCh37/hg19
[c]Based on information provided by the Illumina manifest
[d]Number of EPIC probes mapping close to the DMPs are listed and whether these probes have a p < 0.01 (two-sided Wilcoxon signed-rank test)
[e]Behaves like a methylation quantitative trait loci

is located on the X-chromosome (chrX:130863481–130863509) and encodes a circular, long, non-coding RNA (Supplementary Figs. 9 and 10)[46]. This MS-DMR is not covered by the EPIC array, but a probe (cg08117231) located 6 bp upstream of this DMR was not significant in the PBMC-based EWAS, nor in the females-only analysis (p > 0.05, Wilcoxon signed-rank test). When performing the analysis using a less robust methylation difference of >0.15, then 19 additional MS-DMRs were identified (Supplementary Table 3). Five of these were also covered by the EPIC array, but were not significant in the PBMC-based EWAS. Only 11 of these additional MS-DMRs showed overall consistent methylation differences across the entire DMR, including an MS-DMR in the *DDAH1* gene that also contains an established MS-associated SNP[3]. Unfortunately, the *TMEM232* and *ZBTB16* loci did not fulfill the filtering criterion of ≥10 reads coverage across all samples, but note that they were validated by TDBS.

**Within-pair DMRs are common among MZ twins.** Our PBMC-based EWAS concentrated on the identification of MS-DMPs showing differences across many twin pairs. However, since MS is a heterogeneous disease, DNA methylation changes present in only a few cases should also be considered. Therefore the EPIC array data was used to identify within-pair DMRs (WP-DMRs). To detect robust methylation changes in individual twin pairs, WP-DMRs were defined as ≥3 CpGs within 1 kb having a Δβ-value (adjusted for cell-type composition) > 0.20 and the aberrant methylated co-twin a β-value greater than ±3 standard deviations from the mean. Overall, 45 WP-DMRs were identified in 17 of the 45 twin pairs, ranging from one to 11 WP-DMRs per pair (Supplementary Table 4 and Supplementary Figs. 11–14). Of the 45 WP-DMRs, 43 were solitary and pair-specific and only two WP-DMRs (*ISOC2* and *HIST1H3E*) were found in two independent pairs (Supplementary Fig. 11), but the aberrant methylation pattern did not correlate with the MS phenotype (Supplementary Table 4). Of the 43 pair-specific WP-DMRs, 16 showed an aberrant methylation pattern in the non-affected co-twin and 27 in the MS-affected co-twin. These WP-DMRs have not been associated with MS in other EWAS, nor do they overlap with MS-associated genes reported in the GWAS Catalog (accessed May 2018)[47]. Two pair-specific WP-DMRs were located in reported imprinted DMRs[48] in *SVOPL* and *HM13/MCTS2P* (Supplementary Fig. 12), but in both cases the non-affected co-twin showed an abnormal methylation pattern. Lowering the WP-DMR Δβ-value threshold to 0.15 revealed that of the 27 WP-DMRs, which were aberrantly methylated in the MS-affected

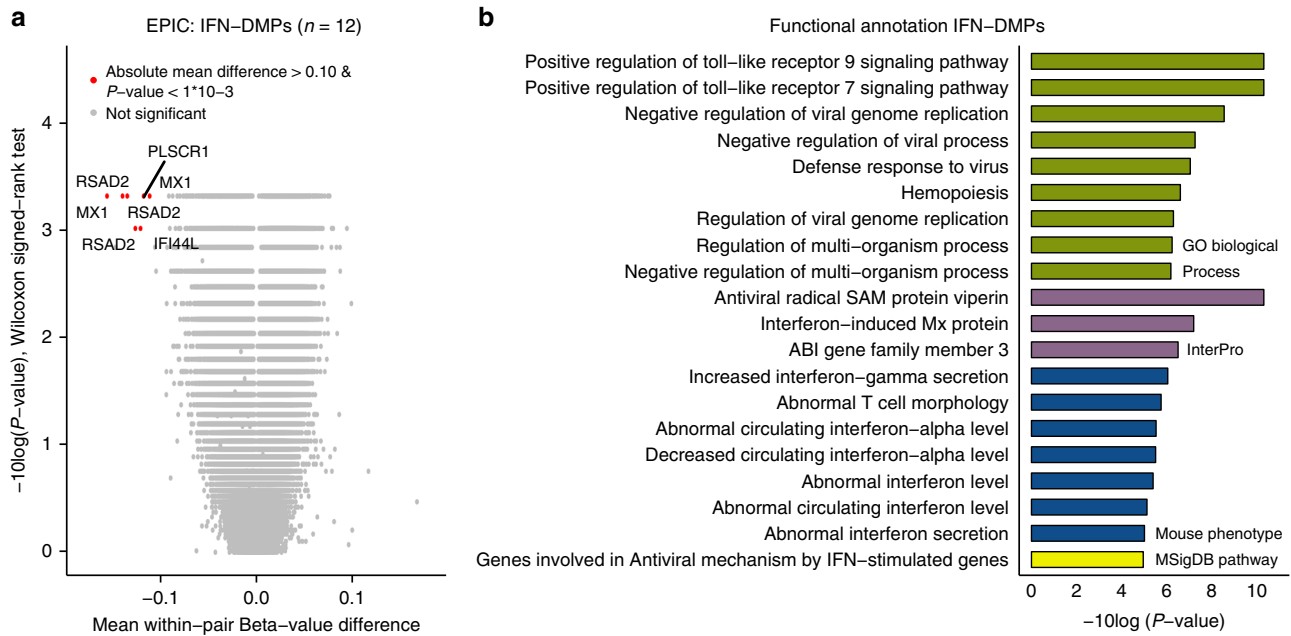

**Fig. 3** Interferon-beta (IFN) treatment-associated DNA methylation changes. **a** Results of the differential DNA methylation analysis including only the EPIC array data of the 12 pairs, of which the MS-affected MZ co-twin was treated with IFN at the moment of blood collection. The volcano plot presents the *p*-values resulting from the nonparametric two-tailed Wilcoxon signed-rank test vs. the mean within-pair β-value difference for each CpG. Data were not adjusted for cell-type composition. Within-pair β-value difference (Δβ-value) = MS-affected IFN-treated MZ co-twin - clinically non-affected MZ co-twin. *n* = number of twin pairs. **b** Summary of the functional annotation analysis using GREAT[42], on the 257 IFN-associated differentially methylated CpG positions (IFN-DMPs) (absolute mean within-pair β-value difference >0.05 and two-sided Wilcoxon signed-rank test *p* < 0.001). Annotation terms are ranked according to their enrichment *p*-values calculated by GREAT[42]: GO Biological Process terms (Hyper raw $p < 1 \times 10^{-6}$) and the other presented terms (Hyper raw $p < 1 \times 10^{-5}$)

co-twins, four WP-DMRs were also identified in other twins. Of these one intergenic WP-DMR was present in four pairs and always hypermethylated in the MS-affected co-twins (Supplementary Table 5). Furthermore, among the 23 pair-specific WP-DMRs, which were aberrantly methylated in the MS-affected co-twins, four WP-DMRs were identified in one pair in the protocadherin gamma (*PCDHG*) gene cluster (Supplementary Fig. 13), and another was observed in the promoter of the non-clustered protocadherin 10 (*PCDH10*) gene (Supplementary Fig. 14). Protocadherins are highly expressed in the brain and involved in neuronal development[49].

**Methylation variability not enhanced in MS-discordant twins.** Increased DNA methylation variability has been observed in MZ twins discordant for the autoimmune diseases type 1 diabetes (T1D) and rheumatoid arthritis (RA)[12,13]. Hence, we tested whether DNA methylation variability is also implicated in MS using the iEVORA algorithm[50]. Applying the default FDR < 0.001 resulted in only 25 differentially variable CpG positions (DVPs) of which the majority (88%) was hypervariable in the non-affected co-twins (Supplementary Table 6 and Supplementary Fig. 15). Hence, our PBMC-based EPIC array data does not support the presence of an MS-associated DNA methylation variability signature in these MS-discordant MZ twins.

**IFN treatment induces robust DNA methylation changes.** Our study design also allows to identify MS treatment-related DNA methylation changes. In our cohort, IFN is the most common disease-modifying treatment, and although IFN-induced transcriptomic alterations in blood cells of MS patients have been studied previously[51–53], DNA methylation changes have not been reported so far. We performed a pair-wise analysis including the EPIC array data of the 12 pairs of which the MS-affected co-twins were treated with IFN at blood collection. The mean Δβ-values

were larger in this subcohort (Fig. 3a), as we identified 257 DMPs with an absolute mean Δβ-value > 0.05 and *p* < 0.001 (Wilcoxon singed-rank test). None of the MS-DMPs listed in Table 2 were among these 257 IFN-associated DMPs (IFN-DMPs). The 257 IFN-DMPs were annotated to 212 genes, of which 124 genes (58%) overlap with IFN-regulated genes recorded in the INTERFEROME gene expression database (accessed May 2018)[51]. Functional annotation analysis revealed clear enrichment for genes involved in antiviral defense and interferon homeostasis (Fig. 3b). Moreover, seven IFN-DMPs had an absolute mean Δβ-value > 0.10 and *p* < 0.001, due to strong hypomethylation in the IFN-treated MS-affected co-twins (Supplementary Table 7 and Supplementary Fig. 16). These seven DMPs were located in *RSAD2* (*n* = 3), *MX1* (*n* = 2), *IFI44L* (*n* = 1) and *PLSCR1* (*n* = 1), i.e., genes reported to be up-regulated in blood cells of IFN-treated MS patients[51–53]. Although the estimated NK and B cells proportions differed significantly between the IFN-treated MS-affected and non-affected co-twins (Supplementary Table 8), adjusting the data for cell-type composition resulted in only a slight attenuation of the IFN-effect (Supplementary Table 7). Hence, our results indicate that these seven DMPs are robust markers for monitoring IFN treatment effects in PBMCs.

**GC treatment induces hypomethylation at *ZBTB16* enhancer DMP.** Among the MS-DMPs, the *ZBTB16* DMP (cg25345365) had the largest effect size (~4%) and is located in an enhancer in intron 3 of *ZBTB16*, which encodes for a transcription factor also known as promyelocytic leukemia zinc finger (PLZF). ZBTB16/PLZF has been reported to be essential for NKT cell development[40], and to contribute to T-helper 17 (Th17) cell differentiation and phenotype maintenance[54]. However, *ZBTB16* is also known as a major GC response gene, being highly upregulated after GC exposure[55], and several days of high-dose intravenous GC therapy is generally used to treat relapses in MS.

None of the MS-affected co-twins included in the array-based EWAS received GCs within three months prior blood collection. Nevertheless, GC treatment constitutes a serious confounder, because 43 of the 45 MS-affected co-twins have a GC treatment history (and the healthy co-twins not). Of these 14 received GCs within >3–12 months prior to blood collection (i.e. high-dose intravenous methylprednisolone (IVMP) 1 g/day for at least 3 days and on average 6 days). In those 14 pairs, the within-pair methylation differences at the *ZBTB16* DMP are significantly larger (more negative) than the pairs in which the MS-affected co-twin had received GCs longer than 12 months ago ($p = 0.0004$, Wilcoxon rank-sum test) (Fig. 4a and Supplementary Fig. 17).

This indicates that the strong association between the *ZBTB16* DMP and the MS phenotype is due to the GC treatment history of the MS-affected co-twins.

**GC-induced DMPs are not widespread in GC-response genes.** Then, the EPIC array data of the 14 pairs of which the MS-affected co-twin had received GCs within >3–12 months prior to blood collection was analyzed to study the effect of recent GC treatment history on the PBMC methylomes (Fig. 4b). 320 potential GC-DMPs had an absolute mean Δβ-value > 0.05 and $P < 0.001$ (Wilcoxon signed-rank test) and were annotated to 279

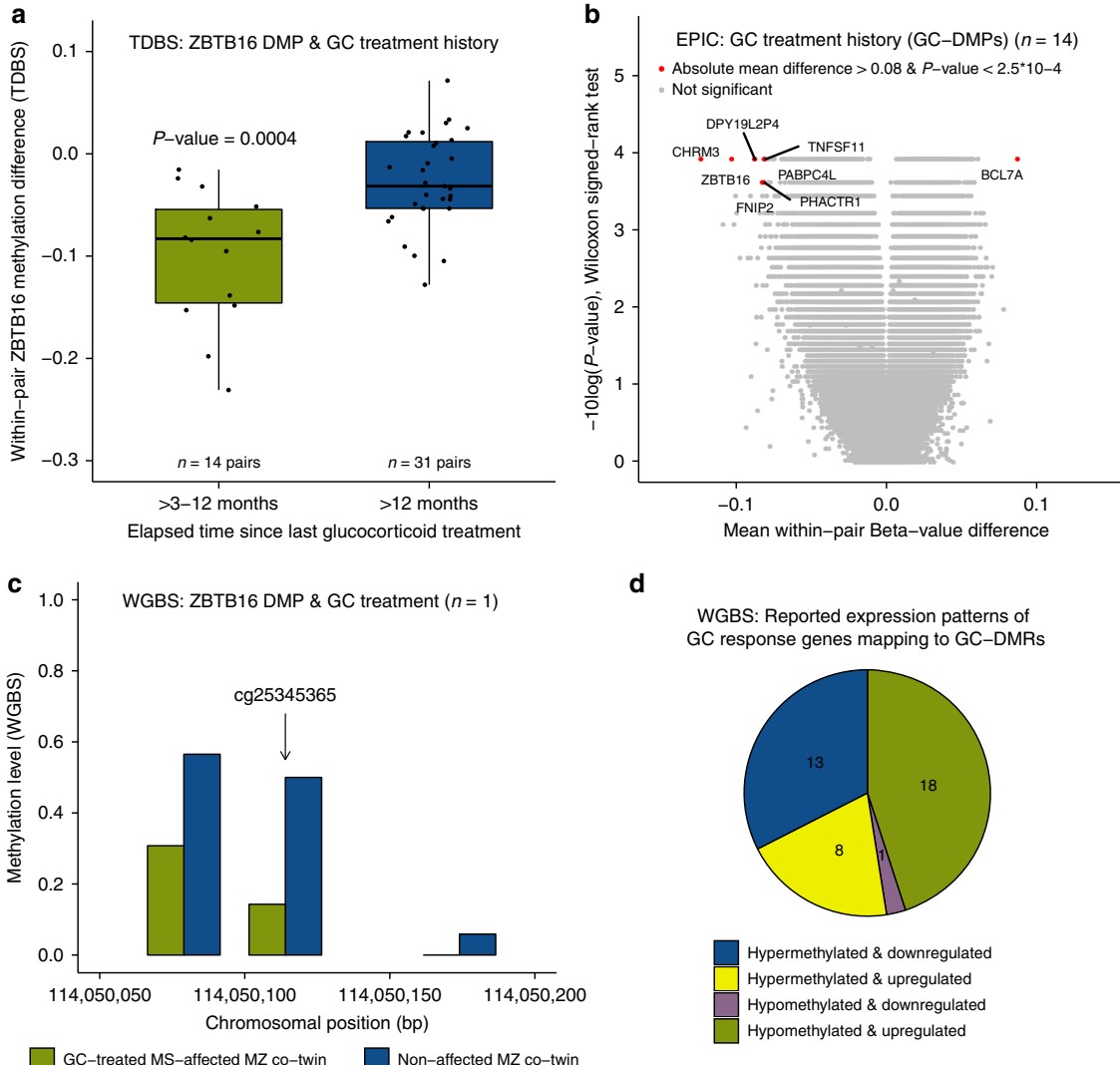

**Fig. 4** Prior glucocorticoid (GC) treatment-associated DNA methylation changes. **a** Within-pair methylation differences of the *ZBTB16* DMP (cg25345365), determined using TDBS in the 14 pairs of which the MS-affected co-twin received GCs within >3–12 months prior blood collection, compared to the 31 pairs of which the MS-affected co-twin was treated with GCs more than 1 year ago. Boxplots represent the median (central line), the interquartile range or IQR (bottom and top of the box), and 1.5 times the IQR (whiskers). *p*-value = nonparametric two-tailed Wilcoxon rank-sum test result. Source data are provided as a Source Data file. **b** Results of the differential DNA methylation analysis including only the EPIC array data of the 14 pairs of which the MS-affected co-twins received GCs >3–12 months prior to blood collection. The volcano plot presents the *p*-values resulting from the nonparametric two-tailed Wilcoxon signed-rank test vs. the mean within-pair β-value difference for each CpG. Data were unadjusted for cell-type composition. **a–b** Within-pair methylation/β-value difference = MS-affected MZ co-twin receiving GCs >3–12 months prior to blood collection – clinically non-affected MZ co-twin. **c** Methylation level of the *ZBTB16* DMP (cg25345365) region determined using WGBS in CD4+ memory T cells of one MS-discordant MZ twin pair of which the MS-affected MZ twin was treated very recently with GCs at the time of blood collection. Coverage at cg25345365 is >20 reads in each co-twin. Source data are provided as a Source Data file. **d** Methylation and reported expression patterns of the 41 GC-DMRs that overlap with GC-response (dexamethasone) genes recorded in the EMBL-EBI Expression Atlas (accessed May 2018). One GC-DMR was excluded because it was reported to be down- and upregulated after dexamethasone treatment (Supplementary Table 9). $n$ = number of twin pairs

genes. Of these, only five genes (1.8%), including *CCNA1*, *GMPR*, *ITGA6*, *LSP1*, and *ZBTB16*, overlap with the 721 GC-response (dexamethasone) genes recorded in the EMBL-EBI Expression Atlas (accessed May 2018). The other four MS-DMPs listed in Table 2 were not among these 320 GC-DMPs.

To study acute GC treatment effects, a WGBS analysis was performed on CD4+ memory T cells of a twin pair of which the MS-affected co-twin was very recently treated with two courses of GCs (i.e., 2 months and 10 days before blood collection with IVMP 1 g/day for 3 and 5 days, respectively). The WGBS data confirmed the strong hypomethylation of the *ZBTB16* DMP (cg25345365) in the GC-treated MS-affected co-twin (36% methylation difference) (Fig. 4c). In addition, 1424 other potential GC-DMRs were identified in the WGBS data, consisting of at least 2 CpGs, absolute mean methylation difference >0.25 and $p < 0.01$ (Wald test). These GC-DMRs were annotated to 682 genes. Only 41 GC-DMRs overlap with 39 (5.7%) GC-response genes reported in the EMBL-EBI Expression Atlas (Supplementary Table 9), which represent potential GC-treatment epigenetic biomarkers. The majority of these 41 GC-DMRs were hypomethylated and the corresponding GC-response gene was recorded as upregulated due to GC treatment (Fig. 4d).

**ZBTB16 methylation and EPIC array-wide hypermethylation**. DNA methylation of the repetitive elements *Alu*, human endogenous retrovirus type K (*HERVK*), and the long interspersed nuclear element-1 (*LINE1*) in the PBMC-derived samples were assessed by TDBS. Although *Alu* methylation correlated significantly with *LINE1* methylation ($r_{Pearson} = 0.43$, $p = 0.003$), methylation levels were overall very similar, showing maximum absolute within-pair differences for Alu, *HERVK*, and *LINE1* of only 0.015, 0.024, and 0.025, respectively. Hence, *Alu*, *HERVK* and *LINE1* methylation levels did not differ between the MS-discordant co-twins ($p > 0.05$, Wilcoxon signed-rank test). Although *Alu* and *HERVK* methylation were affected by cell-type composition differences, adjusting for cell-type composition did not change the results. For Alu generic primers were used, and since the element has ~1 million copies/genome, with a minimum sequencing coverage of 2000 reads/sample about 0.2% of the elements were analyzed. In contrast, *HERVK* and *LINE1* primers were designed to amplify the youngest subfamilies, which gives with a minimum sequencing coverage of 2000 reads >6 fold coverage per individual *HERVK* and *LINE1* element (see legend of Supplementary Table 12).

The volcano plots of the EPIC array data in Supplementary Fig. 2a and Fig. 2 are slightly unbalanced because 59.1% and 55.9% of the CpGs have a positive mean within-pair $β$-value difference before and after cell-type adjustment, respectively. Hence, the EPIC array data suggest an overall hypermethylation in the MS-affected co-twins (Supplementary Table 10). Although this EPIC array-wide hypermethylation in the MS-affected co-twins was not significantly associated with GC treatment history (Supplementary Fig. 18a), the number of hypermethylated CpGs in the MS-affected co-twins correlated significantly with the within-pair *ZBTB16* DMP methylation differences ($r_{Pearson} = -0.36$, $p = 0.02$) (Supplementary Fig. 18b).

**No evidence for within-pair copy-number variations**. Finally, discordant phenotypes within MZ twins can also be due to within-pair copy-number variations (WP-CNVs)[56]. Hence, we checked the EPIC array data for CNVs, but within the MZ twin pairs no chromosomal gains and losses were observed (Supplementary Fig. 19).

## Discussion
Here, we present the largest EWAS in MZ twins clinically discordant for MS to date. Although the PBMC-based methylomes

of the 45 MS-discordant MZ twins were highly similar (mean $Δβ$-values < 0.05), a few new MS-associated candidate loci were identified.

The most prominent MS-DMP was the technically replicated cg25345365 DMP in *ZBTB16*, which has thus far not been reported, probably because the 450 K array used in other MS EWAS studies does not cover this CpG[26–31]. The transcription factor *ZBTB16* is a GC-response gene that becomes highly upregulated after GC exposure[55], and we show that the strong association between the *ZBTB16* DMP and MS in our EWAS is due to GC treatment history. Since none of the MS-affected co-twins had received GCs within three months prior to blood collection, our results indicate that for epigenomic and transcriptomic studies in MS a more stringent inclusion criterion is required (Supplementary Fig. 17). Our results might also have broader implications, because GCs are used in a variety of inflammatory and autoimmune diseases, but dosage and administration route vary per disorder. The GC-glucocorticoid receptor (GCR) complex regulates transcription by binding to glucocorticoid-response elements (GREs) in the genome[57]. GC-GCR binding has been associated with DNA demethylation at enhancer elements, supposedly due to active demethylation[58]. Indeed the *ZBTB16* DMP is hypomethylated in the (GC-treated) MS-affected co-twins, and is located in an enhancer and flanked (<100 bp) by two consensus GRE downstream half-sites (TGTTCT) (Supplementary Fig. 20), which are believed to be sufficient for binding of the GC-GCR complex[57]. In contrast to IFN, a strong GC signature was not observed, because the EPIC array and the WGBS data did not show methylation differences in other common, GC-regulated genes, like *FKBP5* and *TXNIP*[55,59]. However, IFN treatment was ongoing, while GC treatment had been given >3–12 months prior to blood collection. In the WGBS analysis a very recently GC-treated MS-affected co-twin was included, but as this concerned a single-replicate experiment, very stringent analyses criteria had to be applied (e.g. coverage threshold ≥15 reads). Hence, our data reveals the *ZBTB16* DMP as a prominent epigenetic biomarker for GC treatment, and future studies should assess its utility in predicting clinical GC response in patients with inflammatory or autoimmune diseases receiving GC therapy.

Our PBMC-based EWAS also revealed a DMR enriched for MS-DMPs in the *TMEM232* promoter region, which shows enrichment for the chromatin activation mark H3K4me3 in different immune cell types (Supplementary Fig. 21). Despite the small effect size (mean $Δβ$-value = 0.024), this MS-DMR was technically replicated using TDBS, indicating a true effect. *TMEM232* MS-DMPs were also strongly associated with long-standing MS, and no evidence indicated that the association is confounded by treatment history. In whole blood-based case-control 450 K data, no significant difference was observed, but the two most prominent *TMEM232* MS-DMPs were not present on the 450 K array. Nevertheless, neighboring CpGs present on the 450 K array confirmed the directionality of the effect, which might indicate that the MS-DMR is restricted to a PBMC subtype and is diluted in whole blood, in which neutrophils are the predominant cell type. *TMEM232* is a member of the transmembrane (TMEM) protein family, which is predicted to be part of mitochondrial, endoplasmic reticulum, lysosome, and Golgi apparatus membranes[60]. While the function of *TMEM232* is still unknown, variants in this gene have been associated with atopic dermatitis and allergic rhinitis in GWAS[36,37]. Although this might point towards a common immunologic pathway involving *TMEM232*, robust evidence that supports an association between atopic diseases and MS is lacking[61]. Further studies in PBMCs and sorted immune cells are needed to verify the association between the *TMEM232* DMR and MS.

We also carried out a WGBS analysis on CD4+ memory T cells of four MS-discordant female MZ twins. Although this pilot did not reveal widespread global or site-specific MS-associated methylation differences, one potential MS-DMR was identified in an intronic regulatory region in the X-linked *FIRRE* gene. This encodes for a circular, long, non-coding RNA reported to be involved in positioning the inactive X-chromosome to the nucleolus and to maintain histone H3K27me3 methylation[46,62]. While the CpGs within this MS-DMR are not covered by the EPIC array, a probe located 6 bp upstream of the DMR was not significant in the PBMC-based EWAS. This might indicate that this MS-DMR is CD4+ T cell-specific, but can also be the result of stochastic variation caused, e.g., by molecular processes such as X-inactivation. Although our results are preliminary, MS is more common in women[3] and a role of X-inactivation in the pathogenesis of MS has been proposed (reviewed by Brooks et al.[63]); therefore, this DMR represents a possible candidate.

Since MS is a heterogeneous disease, DNA methylation changes present in only a few patients might also contribute to disease manifestation. To identify such rare methylation differences, a WP-DMR analysis was performed, revealing 45 WP-DMRs in 17 twin pairs. This suggests that WP-DMRs are quite common among MZ twins, but, as our analysis is restricted to disease-discordant MZ twins, this cannot be extrapolated to healthy MZ twins. Additional filtering revealed that 24 of these WP-DMRs were associated with the MS phenotype, of which 23 were pair-specific and one intergenic WP-DMR was present in 4 twin pairs. Although these WP-DMRs have not previously been associated with MS, two WP-DMRs were related to genes encoding proto-cadherins that are involved in neuronal development[49]. Hence, a contribution of these WP-DMRs to the discordant phenotype cannot be excluded, but since they are mainly pair-specific, these results should be interpreted very cautiously.

Several observations suggest a role of genomic imprinting in the etiology of MS[18]. In this context, MZ twins are of particular interest because MZ twins discordant for imprinting defects have been described relatively frequently[7–9]. Although we detected two WP-DMRs in reported imprinted DMRs (*SVOPL* and *HM13/MCTS2P*), the aberrant methylation profile was observed in the non-affected co-twin in both cases. Consequently, our PBMC-based analysis does not support the hypothesis that genomic imprinting errors contribute to the discordant clinical manifestation of MS in these MZ twins.

Neven et al.[64] reported hypermethylation of the repetitive elements *Alu*, *LINE1*, and *Sat-α* in blood of MS patients. We also assessed methylation of the repetitive elements *Alu*, *HERVK*, and *LINE1* but observed no differences. However, our EPIC array data does suggest a slight hypermethylation in the MS-affected co-twins. Also Bos et al.[26] observed in their 450 K data evidence for hypermethylation in CD8+ T cells of MS patients, but not for CD4+ T cells or whole blood. While GC treatment history was not directly associated with EPIC array-wide hypermethylation, we observed a rather weak, but significant association between increased within-pair *ZBTB16* methylation differences and the number of hypermethylated CpGs in the MS-affected co-twins. Although further confirmation is needed, this association might indirectly indicate that GCs also affect global DNA methylation levels. This might also explain the strong repetitive element hypermethylation in MS patients reported by Neven et al.[64], who applied an inclusion criterion of only >1 month after GC treatment. However, in our study, hypermethylation in the MS-affected co-twins was only observed in the EPIC array data, and repetitive elements are strongly underrepresented on this array. Accordingly, additional studies are warranted to assess the association between DNA hypermethylation and MS and whether it is confounded by GC treatment history.

All MS-DMPs observed in our study have remained undetected in previous MS EWAS studies[26–31]. However, those studies observed much larger methylation differences and applied absolute mean β-value difference thresholds of >0.05[26] or >0.10[27–30]. As those studies used genetically unmatched cases and controls[26–31], the reported large methylation differences might mainly be driven by genetic variation. Hannon et al.[65] recently showed that, in particular, sites with variable DNA methylation levels and sites robustly associated with environmental exposures are influenced by genetic effects, highlighting the need to control for genetic background in EWAS. Although our discordant MZ twin design perfectly controls for genetic variation, there are also limitations. MS-discordant MZ twins are scarce and therefore it is not possible to control for treatment effects without losing statistical power. Furthermore, the healthy co-twins are at risk to develop MS in the future and subsequently some of the pairs included in this EWAS will get clinically concordant for MS. Since the evolution of MS is supposed to be a continuum it is likely that prior to the clinical onset there is a prodromal phase of undefined duration, with subclinical subtle changes in CSF or MRI pointing to latent neuroinflammation. However, the onset of this postulated prodromal phase is impossible to define and, therefore, we aimed to identify DNA methylation differences that contribute to the discordant clinical manifestation of MS in MZ twins. For the EPIC array analysis, only DNA extracted from PBMCs was available, although it might be more informative to profile distinctive subtypes, such as CD4+ T cells, CD8+ T cells and B cells that are believed to be involved in the pathophysiology of MS[45,66]. Nevertheless, Paul et al.[12] profiled CD4+ T cells, B cells, and monocytes of 50 T1D-discordant MZ twins using the 450 K array, and observed only one genome-wide significant DMP in T cells (mean $\Delta\beta$-value = 0.023)[12]. This might indicate that for detecting robust MS-DMPs even rarer subpopulations such as Th1, Th17, and regulatory T cells need to be profiled, or immune cells in the cerebrospinal fluid. In contrast, Paul et al.[12] identified 10,548 differentially variable CpG positions (DVPs) in B cells, 4314 in CD4+ T cells, and 6508 in monocytes, and the T1D-affected MZ co-twins were enriched for DVPs. In addition, Webster et al.[13] identified 1107 DVPs in whole-blood 450 K data of 79 RA-discordant MZ twins, of which 763 DVPs were hypervariable in the RA-affected MZ co-twins. Although, we used the same method and significance threshold as applied in these studies, we only identified 25 DVPs of which the majority was hypervariable in the non-affected co-twins. Hence, our PBMC-based EPIC data does not reveal an MS-associated DNA methylation variability signature.

In conclusion, our EWAS shows that PBMC-based methylomes of MS-discordant MZ twins are highly similar, and no evidence was found that genomic imprinting errors or CNVs explain the discordant phenotype. However, a few candidate loci were identified, including a MS-DMR in the *TMEM232* promoter. Furthermore, epigenetic biomarkers for MS treatments were identified, revealing that besides short-term also medium-term treatment effects are detectable in blood cells, which should be considered in epigenomic and transcriptomic studies. Overall, we believe that this study represents an important first step in elucidating epigenetic mechanisms underlying the pathogenesis of MS.

## Methods

**Participants**. Twins were recruited by launching a nationally televised appeal and internet notification in Germany with support from the German Multiple Sclerosis Society (DMSG, regional and national division). Inclusion criteria for study participation were met for MZ twins with an MS diagnosis according to the revised McDonald criteria or clinically isolated syndrome in one co-twin only[67]. In total, 55 MZ twin pairs visited the outpatient department at the Institute of Clinical Neuroimmunology in Munich for a detailed interview and neurological examination. To confirm MS diagnosis, medical records including MRI scans from the

patients' treating neurologists were obtained and reviewed. For inclusion in the present analysis, PBMCs had to be available from both co-twins, resulting in 46 MZ twin pairs. The pair that carries the Leber's hereditary optic neuropathy-specific mutation m.11778G>A was not included in this analysis[6]. At blood collection, 23 MS-affected co-twins were receiving disease-modifying treatments, including interferon-beta (IFN, $n = 12$), natalizumab ($n = 5$), glatiramer acetate ($n = 3$), teriflunomide ($n = 1$), and dimethyl fumarate ($n = 2$). None of the MS-affected co-twins included in the array-based EWAS received GCs within three months prior to blood collection. The non-affected co-twins underwent a detailed interview, including a comprehensive history of past and present complaints. In addition, non-affected co-twins were asked in detail for any occurrence of neurological symptoms in the past and an experienced MS neurologist (LAG) performed a neurological examination, including the EDSS. All previous patient registry information, including MRI scans if existing, were obtained and critically reviewed. The study was approved by the local ethics committee of the Ludwig Maximilians University of Munich and all participants gave written informed consent.

**DNA extraction and zygosity determination.** PBMCs were isolated from whole blood using Ficoll density gradient centrifugation and DNA was extracted using the QIAamp DNA Blood Midi Kit (Qiagen, Hilden, Germany). Extracted DNA was treated with RNase A/T1 Mix (Thermo Scientific, Oberhausen, Germany) and subsequently purified using the Genomic DNA Clean & Concentrator™−10 Kit (Zymo Research, CA, USA). As previously described[6], zygosity was confirmed by genotyping 17 highly polymorphic microsatellite markers and by next-generation sequencing of 33 SNPs.

**Infinium MethylationEPIC BeadChip assay.** Genomic DNA was treated with bisulfite using the EZ DNA Methylation kit (D5002, Zymo Research), of which a detailed description is provided in the Supplementary Methods. Both members of a twin pair were always processed in the same batch. Genome-wide DNA methylation profiles of 46 MZ twin pairs clinically discordant for MS were generated using Illumina's Infinium MethylationEPIC BeadChip assay (EPIC array) (Illumina, San Diego, CA, USA) at the Department of Psychiatry and Psychotherapy of the Saarland University Hospital. The assay determines DNA methylation levels at >850,000 CpG sites and provides coverage of CpG islands, RefSeq genes, ENCODE open chromatin, ENCODE transcription factor-binding sites, and FANTOM5 enhancers. The assay was performed according to the manufacturer's instructions and scanned on an Illumina HiScan. To avoid batch effects, both members of a twin pair were always assayed on the same array.

**EPIC array data processing and DMP identification.** Raw EPIC array data were preprocessed using the RnBeads R/Bioconductor package[68]. Low-quality samples and probes were removed using the Greedycut algorithm, based on a detection $p$-value threshold of 0.05, as implemented in the RnBeads package. In addition, probes with less than three beads and probes with a missing value in at least 5% of the samples were removed. For each CpG site, a β-value was calculated, which represents the fraction of methylated cytosines at that particular CpG site ($0 =$ unmethylated, $1 =$ fully methylated). Subsequently, β-values were normalized using Illumina's default normalization method. In total, methylation data of 849,832 sites (866,895 in total) were available for 45 MS-discordant MZ twins. The relatively large number of excluded probes is due to inclusion of early access EPIC arrays, which have 11,652 fewer probes than the final release EPIC arrays. The EPIC array includes 59 SNP sites, which were used for quality control. All MZ twin pairs, except one, shared the same genotypes. The exceptional pair showed only a discordant genotype for SNP rs6471533. However, validation using targeted deep sequencing (TDS) revealed that both co-twins have the same genotype for the rs6471533 SNP (Supplementary Fig. 22), which indicates a technical artifact in the corresponding EPIC probe rather than a true genetic difference.

To identify differentially methylated CpG positions (DMPs) a two-sided non-parametric Wilcoxon signed-rank test was carried out. For the MS EWAS, an arbitrary significance level $\alpha < 5 \times 10^{-6}$ was considered suggestive and genome-wide significance was defined as false discovery rate (FDR) < 0.05. All statistical analyses were performed in R. A functional annotation analysis was performed using the Genomic Regions Enrichment of Annotations Tool (GREAT v3.0.0) with default settings and the EPIC array CpGs, which passed quality control, as background[42].

**Power calculation.** Since the power function of the Wilcoxon signed-rank test is difficult to express[69], we used its closest parametric equivalent (paired T-test) to estimate the power of our MS EWAS. With a sample size of 45 twin pairs, >98% power is achieved to detect a mean β-value difference of at least 0.05 with a (genome-wide) significance threshold of $1 \times 10^{-7}$, using a two-sided paired T-test and assuming a standard deviation of 0.0266 (which is the true median standard deviation observed in our data). Details of this power calculation and calculations using smaller mean β-value differences are presented in Supplementary Table 11. The power analysis was performed using SAS University Edition.

**Estimation of cell-type composition.** A detailed description of the cell-type composition estimation is provided in the Supplementary Methods. In brief, the

cell-type composition of each PBMC sample was estimated using the Houseman algorithm implemented in the *minfi* R/Bioconductor package[70]. The obtained *minfi* estimates were used to adjust the β-values for cellular composition using linear regression and the residuals were used for downstream analysis. To obtain interpretable, adjusted β-values, the unadjusted mean β-value of each CpG site was added to the residuals. To check the quality of the adjustment, the adjusted β-values were used to recalculate the within-pair correlations. As a result, Supplementary Fig. 23 shows that the overall within-pair correlations are, as expected, higher after adjusting for cell-type composition.

**Within-pair DMR analysis.** To identify WP-DMRs in the EPIC array data, the β-value differences (Δβ-values) (adjusted for cell-type composition) per CpG were calculated for each twin pair (the 257 IFN-associated CpGs were excluded). To avoid false positives caused by single probes, WP-DMRs were defined as ≥3 CpGs, each having an absolute Δβ-value > 0.2 with a maximum 1 kb distance between neighboring CpGs. To exclude regions that are characterized by overall variable methylation levels, WP-DMRs were only considered when the β-value of the aberrant methylated co-twin was more than three standard deviations away from the mean.

**DVP identification.** For the DVP analysis, probes containing a SNP within five bases of the measured CpG site, probes mapping to the sex chromosomes, and probes with at least one missing value were excluded, resulting in methylation data of 759,291 sites. DVPs were identified using the iEVORA algorithm[50], which measures differential variability between two groups by utilizing the Bartlett's test to detect differences in variance and an unpaired T-test to identify difference in means. CpGs with a FDR-corrected Barlett's $p < 0.001$ and raw $t$-test $p < 0.05$ were defined as DVPs.

**Copy-number variation analysis.** CNV analysis with the EPIC array data was performed using the *conumee* R/Bioconductor package with default settings (http://bioconductor.org/packages/conumee/, R package version 1.6.0, Accessed 1 Nov 2016). Individual profiles and output were manually assessed. To define chromosomal gains and losses within the MZ twin pairs, an absolute segment mean threshold ≥0.3 was applied.

**Targeted deep bisulfite sequencing.** TDBS was used to validate DMPs resulting from the EPIC array analysis and to determine methylation levels of the repetitive elements *HERVK*, *LINE1*, and *Alu*. Amplicons were generated on bisulfite-treated DNA using region-specific primers with TruSeq adaptor sequences on their 5′-ends (Illumina). Reaction conditions and primer sequences are described in Supplementary Table 12. Purified PCR products were quantified, pooled, amplified using index primers (five cycles), and sequenced in a 300-bp paired-end MiSeq run (Illumina). After demultiplexing, adaptor trimming, and clipping overlapping mates, the resulting FASTQ files were imported into BiQ Analyzer HiMod[71] to filter out low-quality reads and call the methylation levels. Final coverage was >1500 reads/base.

**Targeted deep sequencing.** The rs6471533 SNP was genotyped using TDS (see Supplementary Table 12 for reaction conditions and primer sequences). The workflow is similar to that described for TDBS, except that genomic DNA was used and that the resulting FASTQ files were aligned to the reference sequence using Bowtie 2. Subsequently, variants were called with SAMtools mpileup and variant information was extracted using filter pileup. Final coverage was >1500 reads/base.

**Third-party MS case–control cohort data analysis.** The whole blood-based 450 K data of 140 unrelated MS patients and 139 controls from Kular et al.[31] (GSE106648) were available as intensity matrices of methylated and unmethylated probe intensities, which were imported into R using RnBeads[68]. As no quality information was provided, Greedycut and bead-based filtering was not possible. Proportions of major cell types were estimated as described above. In line with the original study[31], MS-DMPs were determined using linear regression with limma[72] as implemented in RnBeads adjusting for gender, age, smoking status, and the first two principal components of the estimated cell proportion matrix. Adjustment for the batch effect was not possible as the corresponding variable was not provided.

**Whole-genome bisulfite sequencing in CD4+ T cells.** WGBS was used to profile CD4+ central and effector memory T cells of four MS-discordant female MZ twin pairs (mean age 43.3 years, discordant for MS > 12 years, Supplementary Table 13). Of one pair, the MS-affected co-twin had been treated very recently with GCs at the time of blood collection (but never received any immune-modulating therapy), while the MS-affected co-twins of the other three pairs had not received GCs or other immune-modulating therapies within at least 12 months prior to blood collection. The cell sorting procedure, the preparation of the WGBS libraries, and the preprocessing of the WGBS sequencing data are described in detail in the Supplementary Methods. The coverage statistics of the samples are summarized in Supplementary Table 13.

**DMR identification in the WGBS data**. To identify MS-associated DMRs (MS-DMRs), the WGBS data of all four pairs were analyzed using the RnBeads package, in which a paired $t$-test was performed for every CpG. Only CpGs with a coverage ≥10 reads in all samples were included, resulting in methylation information of about 2.7 million CpGs (Supplementary Table 13). MS-DMRs were defined as ≥3 CpGs, each having $p < 0.05$ (two-sided paired $t$-test) and an absolute mean methylation difference >0.2, and a maximum of 500 bp distance between neighboring significant CpGs.

To identify GC treatment-associated DMRs (GC-DMRs), the WGBS data of the pair with the GC-treated MS-affected co-twin were analyzed using DSS-single[73], which is designed to detect DMRs from WGBS data without replicates. To increase the quality of this single-replicate DMR analysis, only CpGs with a coverage ≥15 reads in both samples were included and the sex chromosomes were excluded, resulting in methylation data of up to 2.8 million CpGs (Supplementary Table 13). It has been reported that binding of the GCR complex is rare within CpG islands and predominantly occurs at distal regulatory elements[58]. To detect DMRs in such CpG poor regions, the DSS settings included a smoothing span of 100 bp and minimum DMR length of 25 bp with ≥ 2 CpGs and $p < 0.01$ (Wald test). The absolute mean methylation difference had to be larger than 0.25, and to limit the number of false positives only GC-DMRs located in reported GC-response genes were considered. The GC and MS-DMRs were annotated using the ChIPseeker R/Bioconductor package (v1.14.2)[74].

**Partially methylated domain analysis in WGBS data**. The WGBS data was segmented into PMDs, low methylated regions (LMRs), and unmethylated regions (UMRs) using the MethylSeekR R/Bioconductor package[75]. After filtering gaps annotated by UCSC, the rest of the genome was designated as fully methylated regions (FMRs). As input for MethylSeekR the aggregated strand information per CpG was used, and the MethylSeekR settings included coverage ≥5 reads per CpG, 50% methylation and an FDR ≤ 0.05 for calling hypomethylated regions, resulting in a cut-off of ≥ 4 CpGs per LMR. For each segment, the methylation levels between the non-affected and MS-affected co-twins were compared using a paired t-test on the median weighted average methylation values. In addition, to assess the genome-wide PMD similarity across the eight samples, the genome was binned into 1 kb windows and each was annotated with 1 if the bin overlapped with a PMD and with 0 otherwise. Based on this binarized matrix a hierarchical clustering was performed in R using ward.D2 as agglomeration method and euclidean as a distance measurement. The very same procedure was performed for FMRs.

**Reporting summary**. Further information on research design is available in the Nature Research Reporting Summary linked to this article.

## Data availability

The epigenomic data has been deposited at the European Genome-phenome Archive (EGA, http://www.ebi.ac.uk/ega/), which is hosted at the EBI, under accession number EGAS00001003147. The source data underlying Table 2, Supplementary Tables 1–9, Figs. 2c, 4a, c, and Supplementary Figs. 5, 6, 9, and 11–18 are provided as a Source Data file. The authors declare that all other data are contained within the article and its supplementary files or available from the author upon request.

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

## Acknowledgements

We are grateful to all the twins, who participated in this study. We thank Jasmin Gries for performing MiSeq sequencing, Karl Nordström for preprocessing the reads, Kathrin Kattler for assisting tWGBS protocol optimization, and Katja Anslinger for zygosity determination. We acknowledge the use of the CF FlowCyt at the Biomedical Center of the Ludwig-Maximilians-Universität München. This work was supported by the Gemeinnützige Hertie Stiftung; German MS Foundation (regional and national division); German Research Council [SFB-TR 128 SyNergy]; Krankheitsbezogenes Kompetenznetz Multiple Sklerose, Cyliax Stiftung, Verein zur Therapieforschung für MS Kranke, and the German Federal Ministry of Education and Research (BMBF) funded program DEEP (01KU1216F).

## Author contributions

Study design: N.Y.S., L.A.G., T.K., R.H., J.W. Patient recruitment and care: L.A.G., T.K. Clinical data collection: L.A.G. Experimental work and data collection: N.Y.S., G.G., E.B. Data processing: N.Y.S., P.L. Data analysis: N.Y.S., P.L., A.S. Supervising data processing and analysis: P.L. Facilitating technical and material support: C.P., D.W., J.W. Supervision: R.H., J.W. Manuscript writing: N.Y.S. Manuscript editing: all authors.

## Additional information

**Competing interests:** The authors declare no competing interests.

