## [Peer Review File · Nature Communications]

Reviewer #1 (Remarks to the Author):

Review of 'DNA methylation signatures of a large cohort monozygotic twins clinically discordant for multiple sclerosis'

General comments

In this manuscript, the authors have presented an interrogation of the methylation landscape of 45 multiple sclerosis (MS) discordant monozygotic twin pairs. They identify methylation differences associated generally with MS, and additionally they identify methylation signatures which characterise the effect of Interferon and Glucocorticoid therapy in MS. They also identify within pair differentially methylated regions and report no occurrence of copy number variation within twin pairs.

This is a well-designed study, which addresses the confounding effect of genetic heterogeneity which affects many population-wide epigenome wide association studies of common disease by investigating disease-discordant monozygotic twin pairs. The initial results of MS-related differentially methylated positions are somewhat overstated, with adjusted P values of ~ 0.74 following adjustment for cell composition effects. However, the methylation differences identified relating to the effects of treatment of DNA methylation are much more convincing and could potentially have important implications for our knowledge of the action of these therapies, which could inform treatment strategy in future, following further research.

While the manuscript is generally quite well written, grammar needs to be improved and references used in the introduction should be updated as they do not reflect recent advances in the field of DNA methylation research.

Specific points

The approach taken to validate methylation measurements from the array using targeted deep bisulphite sequencing is a sensible one, however the two CpG sites selected for validation using targeted deep bisulphite sequencing do not reach the significance threshold following adjustment for cell composition and have very small methylation differences of around 2%. While it is impressive that these validate using sequencing, it is not enough evidence to draw strong conclusions from without validation in an independent cohort. I appreciate that it is a unique cohort and that finding a replication cohort of disease discordant monozygotic twins would be extremely difficult, but perhaps it could be tested in methylation data from unrelated individuals. The authors should also investigate the degree of DNA methylation variability between affected and unaffected twins as this has recently been implicated in other autoimmune diseases (type 1 diabetes and rheumatoid arthritis).

The identification of methylation changes triggered by treatment with interferon-beta therapy is very interesting, especially as the methylation alterations map to genes regulated by IFN. The predicted cell proportions should also be compared in this subgroup to demonstrate if they are different or not.

In the introduction, the presence of SNP-containing probes on the array is mentioned however this is not addressed in the analysis. Despite investigating monozygotic twins, these analyses could still be confounded by inter-twin pair genetic variation and so probes which contain a SNP within the first 3-5 base pairs from the CpG site of interest should be removed. This can be done in a population-specific manner to reduce the number of probes removed. A comprehensive list, and justification for removal, can be found in Zhou 2017 (Nucleic Acids Research).

It is not clear from the current analysis if the methylation differences identified are dependent on disease duration. The 'years clinically discordant for MS' ranges from 0-45 and the 'age of onset' ranges from 14-46, could the authors comment on the effect of long-standing disease versus recently-diagnosed, and the effect of early vs late onset disease?

The treatment related methylation differences identified appear to gradually return to the 'untreated' state following cessation of glucocorticoid treatment, could the authors comment on the duration of treatment needed to initiate these methylation differences to begin with?

Smoking behaviour is also mentioned in the introduction as it influences DNA methylation and development of MS, however it is not included in patient demographics in Table 1. Was this data available and if so did it affect methylation measurements? If data is not available, smoking status could potentially be predicted from the DNA methylation profiles, using a prediction algorithm such as that described in Elliott 2014 (Clinical Epigenetics).

Regarding the occurrence of within-pair differential methylation between affected and unaffected twins, could the authors comment on the expected degree of methylation variability in healthy monozygotic twin pairs?

Finally, the mean beta value of comparison groups should be included in tables describing differentially methylated positions.

Reviewer #2 (Remarks to the Author):

Souren et al. investigated DNA methylation differences in peripheral blood mononuclear cells (PBMCs) of a large cohort of clinically discordant Multiple Sclerosis (MS) MZ twins. This is a second attempt to characterize methylation differences in discordant twins but this time in a considerably larger cohort comprising 45 discordant twins. It is thus a bit disappointing that no robust differences have been detected. Nevertheless, the study comprises an important contribution to the field describing the status of PBMC methylomes in discordant MS twins, and, importantly, providing the signatures of interferon and glucocorticoid treatments, which need to be taken into account in this type of studies.

Major concerns:

1. How well is the clinical status of the non-affected twin established? Only half of the twins are clinically discordant for more than 10 years and there are some pairs (how many?) that are clinically discordant for 0 years (range of 0-45). Have MRI and CSF analyses been done on unaffected twins; has previous patient registry information been reviewed? This should be explained in detail. What would be the outcome (including correction for appropriate confounders – see below) in a sub-cohort of 25 pairs that were discordant for more than 10 years?
2. Authors correctly state that MS is a highly heterogeneous disease and the collected twin cohort is also highly heterogeneous (in terms of the MS type, duration, severity, treatment, age etc.) raising a possibility for a significant influence of confounders. For example, the age span is 21-67 and age is known to associate with methylation changes and could therefore interact with MS status (similar applies to sex). The cohort also comprises different MS types (RR/SP/PP): more than 30% are progressive MS patients where inflammatory processes in PBMCs might not be any longer dominant. A thorough investigation of the impact of all potential confounders should be performed and relevant covariates introduced in analysis. The authors adjusted the β -values for cellular composition using linear regression and the residuals were used for downstream analysis. In this approach, the other covariates (age, sex, batches etc.) should be included, otherwise cell proportions may delete their effects.
3. WP-DMR analysis is highly relevant and interesting, however, $\Delta\beta$ -value “ > 0.20 and the aberrant methylated co-twin having a β -value greater than ± 3 ” might be too stringent given the heterogeneous cellular source (PBMCs) and the fact that finding differences typical for a subgroup of MS patients (not an individual pair) would be most relevant in terms of interpretability. The less stringent criteria (e.g. lower $\Delta\beta$) should be at least considered when looking for evidence of a WP-DMR in other twin pairs. However, it would be highly desirable to see this analysis where the focus is on finding within a subgroup differences (would likely require less stringent $\Delta\beta$).
4. Have authors attempted to validate FIRRE (which candidate CpG is not present on the EPIC array) in the PBMC cohort of 45 twins with TDBS (or other technique)?

5. Line 260: “slightly unbalanced, because 59.1% and 55.9% of the CpGs have”. What is this balance if $\Delta\beta$ cut-offs are introduced together with more stringent p-values. Does it still favour “predominant” hyper-methylation?
6. Correlation presented in Suppl. Fig. 15B is not very convincing. More evidence should be provided or discussion regarding GC induced hypermethylation revisited.
7. While IFN-DMPs, although not significant, demonstrate a convincing IFN signature (i.e. high overlap with IFN genes and IFN-response related GO terms), the genes associated with GC-DMPs don't seem to overlap with known GC-response signatures. Please comment. Is ZBTB16 methylation reported in other MS studies? Do other studies report IFN-DMPs? One would expect some. Please comment.
8. Line 35: “argues against some previously reported MS-associated epigenetic candidates”. This statement should be removed as it is imprecise (what previously reported MS-associated candidates have been excluded by the findings of this study?) and because the study design is considerably different from previous studies.

Minor issues:

1. Line 63: “Although the molecular mechanisms underlying these associations remain unknown, it is possible that these environmental factors operate by inducing DNA methylation changes.” Please indicate if DNA methylation changes in MS in relation to these factors have been reported.
2. Line 111: Please clarify what is a suggestive P-value $< 5 \times 10^{-6}$ based on.
3. Lines 133-135 “Next, a functional annotation analysis carried out with all MS-DMPs having a $P < 0.001$ in the analysis adjusted for cell-type composition (698 in total) revealed TMEM232 as a prime candidate marker for MS. All other annotation categories were not significant.” Please clarify what do these analysis comprise and refer to. It is not clear.
4. Line 173: “45 WP-DMRs were identified in 17 out of the 45 twins”. Is there something in common for MS of these 17 twins i.e. MS type, disease duration etc.?

5. DMPs in TMEM232 are possibly the only truly MS-associated changes. Can one infer something from regulatory region annotation in primary immune cells? Is there any overlap with other methylation studies in MS (even among suggestive loci with same directionality)?

6. What is the regulatory region annotation/chromatin state of FIRRE in various primary CD4 subsets (Blueprint data)?

7. WGBS in CD4+ cells: what is the rationale for using a mean difference of 20%, which is rather large, especially for analysis in a sorted cell type? Is there more overlap with EPIC data or other published data in CD4 with less robust cut-off for a difference?

8. Line 357: "Consequently, our data does not support a contribution of genomic imprinting errors in the pathogenesis of MS." This is too strong statement considering that specific cell types, additional modifications and mechanisms of imprinting have not been investigated.

Reviewer #3 (Remarks to the Author):

Souren *et al.* have investigated Multiple Sclerosis in Monozygotic (MZ) disease-discordant twin pairs by analysis of the DNA methylation with the Illumina EPIC (850k) arrays in 45 pairs and Whole Genome Bisulphite Sequencing (WGBS) in 4 pairs. The former was in peripheral blood mononuclear cell-derived DNA and the later in isolated CD4+ T cells.

On the whole the study is well designed and analysis is robust – with, as expected in an MZ discordant study, the removal of genetic confounding leading cell-type heterogeneity from blood-derived DNA to be the most considerable factor influencing variability. I have some queries below, which the authors could address, that I think would help improve and qualify the results as presented. In particular, points regarding the targeted sequencing to validate the findings and the assessment of the DNA methylation state of repetitive elements.

Major

- 1) The references given for the identification of discordant DNA methylation profiles for MZ twins in imprinted loci are at least a decade old (Pg 3, line 57) – are there any more recent and robust examples of this instead?
- 2) The authors should include more discussion regarding the age distribution of the twins and years clinically discordant between the pairs. Differences in the disease age-of-onset can be a significant issue in MZ discordant studies. The range of 0 – 45 years for clinically discordance in this study (Table 1) indicates some twins have only recently been diagnosed. Therefore, there is the increased potential for some of these MZ pairs to become concordant over time. The authors should explore how this may have influenced their analysis.
- 3) The study of Baranzini *et al.* is mentioned in the Introduction (pg 2, line 79) - it should be also be included in the text that this study's analysis threshold required a near total reversal of methylation state to identify significant CpGs – that is a change between discordant siblings from $\leq 20\%$ to $\geq 80\%$ methylated., or the reverse.
- 4) The “functional annotation analysis” (pg 5, line 133) should be described in more detail here – no reference to method given or description?
- 5) Could the authors comment on whether stochastic variation in DNA sample and random X inactivation could influence the result found within the *FIRRE* gene?
- 6) The authors state that DNA methylation changes were found in the imprinted genes *SVOPL* and *HM13* (pg 7 line 183). Was this within the Imprinting Control Region (ICR) of either gene?
- 7) Cloning was employed to reduce PCR bias with traditional Sanger Bisulphite sequencing. For the same reason, Targeted Bisulphite sequencing methods have used micro-fluidic or droplet methods to minimise potential bias, with, for example, Raindance¹ or Fluidigm² methodology. The authors should comment on the drawbacks of not using such an approach here. Whilst results appear consistent between array and targeted analysis, the $r=0.8$ – so therefore, not exact, which is potentially indicated by the larger methylation range for the TDBS in Figure S4 *THEM232* cg27037608 for example. This imprecision in the method may be more biased in some CpG locations.
- 8) The use of generic primers to assay the repetitive regions, Alu, HERVK, and LINE1, which are exceeding numerous throughout the genome could lead to stochastic findings unless extreme depth of coverage was attained. There are, for example, >1 million copies of Alu elements throughout the genome. Can the authors comment on the appropriateness of this generic

primer methodology and whether the results were reproducible? Additionally, do the primers preferentially assay a subfamily set of any of the repeat families, due to any sequence variation within the primers?

- 9) Whilst the value of assessment of Global DNA methylation is debated beyond examples of gross cancer tissue differences, the authors should acknowledge they are usually performed on total genomic DNA. Therefore, are strongly driven by the methylation state of the repetitive portion of the genome. The EPIC array still only assays ~3% of the CpGs within the DNA methylome and is bias towards the more unmethylated and less repetitive portion than the true genome-wide picture. Consequently, the authors should acknowledge this shortcoming in any inference of the global DNA methylation state in the disease-related twin.
- 10) The Discussion correctly states that population studies have considerable potential for genetic confounding so that genetic effects will be driving many of these observations – see also recent papers^{3,4} on the extent of this potential genetic confounding, both within array and in sequencing based-studies, respectively. This point regarding genetic confounding in population-based studies should be strongly reiterated.
- 11) The observation that the >3 month cut off for GC treatment (pg 10 line 293) was not sufficient is of significant interest. If this could be further supported/replicated the authors could consider including this in the Abstract.

Minor

- 1) The inclusion of “Large cohort” in the title - The authors should either state the precise number of twin pairs or remove. Size comparators date very quickly, and meta-analyses of discordant twin sets are or will be performed very shortly⁵, even for comparatively rare diseases, as in this case.
 - 2) The Abstract is too vague in a number of places and should be re-written to be precise in its details *i.e.* – ‘a few MS-associated DMPs’, ‘many regions’ – instead state exact numbers.
-
1. Paul, D.S. *et al.* Assessment of RainDrop BS-seq as a method for large-scale, targeted bisulfite sequencing. *Epigenetics* **9**(2014).
 2. Korbie, D. *et al.* Multiplex bisulfite PCR resequencing of clinical FFPE DNA. *Clinical Epigenetics* **7**, 28 (2015).
 3. Hannon, E. *et al.* Characterizing genetic and environmental influences on variable DNA methylation using monozygotic and dizygotic twins. *PLoS Genet* **14**, e1007544 (2018).
 4. Bell, C.G. *et al.* Obligatory and facilitative allelic variation in the DNA methylome within common disease-associated loci. *Nat Commun* **9**, 8 (2018).
 5. Willemsen, G. *et al.* The Concordance and Heritability of Type 2 Diabetes in 34,166 Twin Pairs From International Twin Registers: The Discordant Twin (DISCOTWIN) Consortium. *Twin Res Hum Genet* **18**, 762-71 (2015).

We thank all reviewers for the very detailed evaluation of our manuscript, and the constructive comments and suggestions. The points raised have been very helpful for improving the manuscript. We hope that we sufficiently addressed all the issues and look forward to your responses.

Reviewers' comments

Reviewer #1

General comments

In this manuscript, the authors have presented an interrogation of the methylation landscape of 45 multiple sclerosis (MS) discordant monozygotic twin pairs. They identify methylation differences associated generally with MS, and additionally they identify methylation signatures which characterize the effect of Interferon and Glucocorticoid therapy in MS. They also identify within pair differentially methylated regions and report no occurrence of copy number variation within twin pairs.

This is a well-designed study, which addresses the confounding effect of genetic heterogeneity which affects many population-wide epigenome wide association studies of common disease by investigating disease-discordant monozygotic twin pairs. The initial results of MS-related differentially methylated positions are somewhat overstated, with adjusted P values of ~ 0.74 following adjustment for cell composition effects. However, the methylation differences identified relating to the effects of treatment of DNA methylation are much more convincing and could potentially have important implications for our knowledge of the action of these therapies, which could inform treatment strategy in future, following further research.

While the manuscript is generally quite well written, grammar needs to be improved and references used in the introduction should be updated as they do not reflect recent advances in the field of DNA methylation research.

Answer: This issue has also been raised by reviewer #3 and we updated the references in the introduction (highlighted in yellow). In addition, the final version of the manuscript has been read and edited by a native speaker.

Specific points

1) The approach taken to validate methylation measurements from the array using targeted deep bisulphite sequencing is a sensible one, however the two CpG sites selected for validation using targeted deep bisulphite sequencing do not reach the significance threshold following adjustment for cell composition and have very small methylation differences of around 2%. While it is impressive that these validate using sequencing, it is not enough evidence to draw strong conclusions from without validation in an independent cohort. I appreciate that it is a unique cohort and that finding a replication cohort of disease discordant monozygotic twins would be extremely difficult, but perhaps it could be tested in methylation data from unrelated individuals.

Answer: We evaluated the *TMEM232* MS-DMPs in unrelated MS patients and controls by analyzing recently published blood-based 450K EWAS data of 140 unrelated MS patients and 139 controls from Kular *et al.* (*Nature Communications*, 2018) (450K array does not contain the *ZBTB16* MS-DMP cg25345365). Unfortunately, the cg27037608 and cg26583412 *TMEM232* MS-DMPs are not present on the 450K array, but data from seven neighboring CpGs were available. Although none of these seven CpGs was significantly differentially methylated between the unrelated MS cases and controls ($P > 0.05$), the methylation levels were always higher in the MS patients, confirming the directionality of the effect observed in the twin cohort (see Supplementary Table 2). We added this to the Results section on page 7.

In addition, we added the following to the discussion on page 13:

“In whole blood-based case-control 450K data, no significant difference was observed, but the two most prominent TMEM232 MS-DMPs were not present on the 450K array. Nevertheless, all TMEM232 CpGs present on the 450K array confirmed the directionality of the effect observed in the twins, which

might indicate that the MS-DMR is restricted to a PBMC subtype and is diluted in whole blood, in which neutrophils are the predominant cell type.”

2) The authors should also investigate the degree of DNA methylation variability between affected and unaffected twins as this has recently been implicated in other autoimmune diseases (type 1 diabetes and rheumatoid arthritis).

Answer: Paul et al. (*Nature Communications*, 2016) demonstrated in an impressive EWAS of Type 1 Diabetes (T1D) discordant MZ twins, that the T1D-affected MZ co-twins were enriched for differentially variable CpG positions (DVPs) when compared with their healthy co-twins as well as compared with healthy individuals. Paul et al. performed the DVP analysis on 450K data generated on DNA extracted from three different sorted primary cell types, i.e., B cells, CD4+ T cells, and monocytes, in which they identified 10548, 4314, and 6508 DVPs, respectively. In addition, they report that these DVPs are epigenetic outliers often occurring in individual twin pairs and cell types. Since our EPIC array data is generated on DNA extracted from PBMCs that comprise a heterogeneous cell population and our cohort is diverse regarding prior and current MS treatments, we do not consider our data suitable to conduct such a sensitive DVP analysis. As an alternative, we carried out a within-pair DMR analysis, in which we attempt to identify robust differentially variable methylated regions (≥ 3 CpGs within 1 kb having a $\Delta\beta$ -value > 0.20) that are more likely to have biological consequences and a higher potential diagnostic value. We comment on this in the Discussion on page 16.

3) The identification of methylation changes triggered by treatment with interferon-beta therapy is very interesting, especially as the methylation alterations map to genes regulated by IFN. The predicted cell proportions should also be compared in this subgroup to demonstrate if they are different or not.

Answer: In Supplementary Table 7 we summarized the estimated cell type proportions of the 12 pairs of which the clinically MS-affected MZ co-twin was treated with interferon-beta (IFN) at the moment of blood collection. These results show that estimated proportions of NK and B cells differed significantly between the IFN-treated MS-affected and non-affected co-twins. Nevertheless, adjusting the data for cell-type composition resulted only in a slight attenuation of the IFN-effect. We refer to Supplementary Table 7 in the main text at page 9.

4) In the introduction, the presence of SNP-containing probes on the array is mentioned however this is not addressed in the analysis. Despite investigating monozygotic twins, these analyses could still be confounded by inter-twin pair genetic variation and so probes which contain a SNP within the first 3-5 base pairs from the CpG site of interest should be removed. This can be done in a population-specific manner to reduce the number of probes removed. A comprehensive list, and justification for removal, can be found in Zhou 2017 (Nucleic Acids Research).

Answer: We only performed pair-wise comparisons (i.e., analysing within-pair/intra-pair methylation differences) and therefore our analyses are not confounded by inter-twin pair genetic variation.

5) It is not clear from the current analysis if the methylation differences identified are dependent on disease duration. The ‘years clinically discordant for MS’ ranges from 0-45 and the ‘age of onset’ ranges from 14-46, could the authors comment on the effect of long-standing disease versus recently-diagnosed (A), and the effect of early vs late onset disease (B)?

Answer: A) As requested by the other reviewers as well, we performed a stratified analysis only including the 25 pairs that have been clinically discordant for MS for more than 10 years (see Supplementary Table 1). In the analysis adjusted for cell-type composition two DMPs had a suggestive P-value $< 5 \times 10^{-6}$, one of which is located in the promoter region of the *TACSTD2* gene encoding tumor-associated calcium signal transducer 2 (mean $\Delta\beta$ -value = -0.022). The second DMP is located in the promoter region of the *RCLI* gene (mean $\Delta\beta$ -value = -0.011), which has been linked to depression. For these two long-standing MS-DMPs, EPIC array data of neighboring CpGs were also available (< 500 bp), and two neighboring CpGs of the *TACSTD2* MS-DMP showed a similar trend ($P < 0.10$). In

addition, the *TMEM232* MS-DMPs cg27037608 (mean $\Delta\beta$ -value=0.026, $P=3.2*10^{-5}$) and cg26583412 (mean $\Delta\beta$ -value=0.038, $P=1.8*10^{-5}$) were among the top 15 most significant DMPs associated with long-standing MS. Furthermore, the *ZBTB16* MS-DMP had a mean $\Delta\beta$ -value difference of -0.036 and $P=0.002$ (Supplementary Table 1). These results indicate that the *TMEM232* and *ZBTB16* MS-DMPs are also associated with long-standing MS. We added this to the Results section on page 6 and mention it in the Discussion at page 13.

B) MS usually manifests between 20 and 40 years of age in approximately 70% of the cases, with an average age of onset of ~30 years (Liguori *et al.*, *Neurol Sci*, 2000; O'Connor P, *Neurology*, 2002). Since our twin cohort has an average age of onset of 28 years (Table 1), and contains 7 (16%) cases younger than 20 years and 6 (13%) cases older than 40 years at disease onset, the age of onset in our cohort is within the normal range (we added this info to Supplementary Figure 1). Accordingly, the number of cases in our cohort with early and late disease onset are too small to properly investigate the effect of early *versus* late onset.

6) The treatment related methylation differences identified appear to gradually return to the 'untreated' state following cessation of glucocorticoid treatment, could the authors comment on the duration of treatment needed to initiate these methylation differences to begin with?

Answer: Standard treatment for acute relapses in MS is a course of high-dose of intravenous glucocorticoids (methylprednisolone) with 1 g/day for 3–5 days, and in case neurological symptoms do not fully recover an additional course will be applied. We now added more details on the treatment duration and dose in the paragraph entitled “Glucocorticoid (GC) treatment induces hypomethylation at *ZBTB16* enhancer” (page 10):

“None of the MS-affected co-twins included in the array-based EWAS received GCs within three months prior blood collection, but 43 of the 45 MS-affected co-twins presented with a GC treatment history: here, 14 received GCs within >3-12 months prior to blood collection (i.e., high-dose intravenous methylprednisolone (IVMP) 1 g/day for at least 3 days and on average 6 days).”

We also added more details in the paragraph entitled “GC-induced methylation changes are not widespread in GC-response genes” (page 10):

“To study acute effects of GC treatment, a WGBS analysis was performed on CD4+ memory T cells of one MS-discordant MZ twin pair, of which the MS-affected co-twin was very recently treated with two courses of GCs (i.e., two months and 10 days before blood collection with IVMP 1 g/day for 3 and 5 days, respectively).”

Unfortunately, to precisely determine the treatment duration, which is necessary to initiate the methylation differences, a longitudinal study design is required, that (ideally) includes GC-treatment naive MS patients providing blood samples before, during and after treatment.

7) Smoking behavior is also mentioned in the introduction as it influences DNA methylation and development of MS, however it is not included in patient demographics in Table 1. Was this data available and if so did it affect methylation measurements? If data is not available, smoking status could potentially be predicted from the DNA methylation profiles, using a prediction algorithm such as that described in Elliott 2014 (Clinical Epigenetics).

Answer: Detailed data regarding smoking behavior of the twins at disease onset and at the moment that the samples were collected was available and is now included in Table 1. These data show that there were no significant differences in the number of smokers and the pack-years among the MS-affected and non-affected MZ co-twins at disease onset and at the moment that the samples were collected. Hence, it is unlikely that smoking behavior confounds our analysis. Moreover, the effect of smoking on DNA methylation levels is not within the scope of our study and has been extensively addressed by others. As requested by reviewer 2, we now cite in the introduction two studies that evaluated the effect of smoking on DNA methylation in blood, including a meta-analysis using 450K data of 15907 individuals (Joehanes *et al.*, *Circ Cardiovasc Genet*, 2016).

8) Regarding the occurrence of within-pair differential methylation between affected and unaffected twins, could the authors comment on the expected degree of methylation variability in healthy monozygotic twin pairs?

Answer: Unfortunately, we do not have EPIC array data available of healthy MZ twins, so we cannot verify this, but we added the following statement to the discussion (page 14).

“..., in total, 45 WP-DMRs were identified in 17 twin pairs. This suggests that WP-DMRs are quite common among MZ twins, but, as our analysis is restricted to disease-discordant MZ twins, this cannot be extrapolated to healthy MZ twins.”

9) Finally, the mean beta value of comparison groups should be included in tables describing differentially methylated positions.

Answer: Mean β -values of the MS-affected and non-affected MZ co-twins were added to the respective tables (i.e., Table 2, Supplementary Table 1, 2 and 6, and Supplementary Figure 5 and 6).

Reviewer #2

Souren et al. investigated DNA methylation differences in peripheral blood mononuclear cells (PBMCs) of a large cohort of clinically discordant Multiple Sclerosis (MS) MZ twins. This is a second attempt to characterize methylation differences in discordant twins but this time in a considerably larger cohort comprising 45 discordant twins. It is thus a bit disappointing that no robust differences have been detected. Nevertheless, the study comprises an important contribution to the field describing the status of PBMC methylomes in discordant MS twins, and, importantly, providing the signatures of interferon and glucocorticoid treatments, which need to be taken into account in this type of studies.

Major concerns:

1. A) How well is the clinical status of the non-affected twin established? Only half of the twins are clinically discordant for more than 10 years and there are some pairs (how many?) that are clinically discordant for 0 years (range of 0-45). Have MRI and CSF analyses been done on unaffected twins; has previous patient registry information been reviewed? This should be explained in detail. B) What would be the outcome (including correction for appropriate confounders – see below) in a sub-cohort of 25 pairs that were discordant for more than 10 years?

Answer: Inclusion criteria for study participation was clinical discordance for MS. All healthy co-twins underwent a detailed interview including a comprehensive history of past and present complaints. In addition, healthy co-twins were asked in detail for any occurrence of neurological symptoms in the past and an experienced MS neurologist (LAG) performed a neurological examination, including the EDSS. All previous patient registry information, including MRI scans if existing, were obtained and critically reviewed. We added this to the Methods section at page 17.

At the time point of blood sampling MRI and CSF analysis have been performed in a small portion of healthy-co-twins only and hence this data is not shown. Currently follow-up visits with detailed MRI and CSF analyses are in progress. However, regarding the four twin pairs included in the WGBS analysis of CD4+ T cells MRI and CSF examinations were performed at a follow-up visit and showed unremarkable results with no signs of subclinical neuroinflammation.

The evolution of MS is supposed to be a continuum. It is likely that prior to the clinical onset there is a prodromal phase of undefined duration, with subclinical subtle changes in CSF or MRI pointing to latent neuroinflammation. The onset of this postulated prodromal phase is impossible to define. Hence our study aimed to identify methylation differences associated with a phenotype of clinically definite MS. We are aware, however, that the healthy co-twins are at a high genetic risk to develop MS and therefore we agree that a stratified analysis including only the 25 pairs with a time interval of discordance >10 years is the best practical option to address this problem. We comment on this in the discussion at page 16.

Since the reviewer wonders how many twins are clinically discordant for 0 years, we now included in Table 1 a more accurate characteristic called “Years clinically discordant for MS at sample collection“ which has a range from 1 to 45 years. To give a precise impression of how this variable is distributed, we included in the Supplementary Information a boxplot (including all data points) of the years that the MZ twins were clinically discordant for MS at sample collection (see Supplementary Figure 1B).

B) As requested by the other reviewers as well, we performed a stratified analysis only including the 25 pairs that have been clinically discordant for MS for more than 10 years (see Supplementary Table 1). In the analysis adjusted for cell-type composition, two DMPs had a suggestive P-value < 5×10^{-6} , one of which is located in the promoter region of the *TACSTD2* gene encoding tumor-associated calcium signal transducer 2 (mean $\Delta\beta$ -value = -0.022). The second DMP is located in the promoter region of the *RCL1* gene (mean $\Delta\beta$ -value = -0.011), which has been linked to depression. In addition, the *TMEM232* MS-DMPs cg27037608 (mean $\Delta\beta$ -value = 0.026, P = 3.2×10^{-5}) and cg26583412 (mean $\Delta\beta$ -value = 0.038, P = 1.8×10^{-5}) were among the top 15 most significant DMPs associated with long-standing MS. Furthermore, the *ZBTB16* MS-DMP had a mean $\Delta\beta$ -value difference of -0.036 and P = 0.002 (Supplementary Table 1). These results indicate that the *TMEM232* and *ZBTB16* MS-DMPs are also

associated with long-standing MS. We added this to the Results section on page 6 and mention it in the Discussion at page 13.

2. Authors correctly state that MS is a highly heterogeneous disease and the collected twin cohort is also highly heterogeneous (in terms of the MS type, duration, severity, treatment, age etc.) raising a possibility for a significant influence of confounders. For example, the age span is 21-67 and age is known to associate with methylation changes and could therefore interact with MS status (similar applies to sex). The cohort also comprises different MS types (RR/SP/PP): more than 30% are progressive MS patients where inflammatory processes in PBMCs might not be any longer dominant. A thorough investigation of the impact of all potential confounders should be performed and relevant covariates introduced in analysis. The authors adjusted the β -values for cellular composition using linear regression and the residuals were used for downstream analysis. In this approach, the other covariates (age, sex, batches etc.) should be included, otherwise cell proportions may delete their effects.

Answer: The reviewer correctly states that EWAS studies are often hampered by confounding factors, such as gender and age. However, a certain factor can only be a confounder when it is unequally distributed between the groups being compared. The advantage of an MZ twin design is that they are matched for a range of possible confounding factors, including genetic background, age, gender, and many environmental influences. Since we only perform pair-wise comparisons (i.e., analyzing within-pair methylation differences), variables like age, sex and SNPs do not confound our analyses. In addition, the twin pairs were always processed in the same batch, so batch is also not a confounder in our study. Therefore we do not need to adjust our data for possible confounding effect of these variables. However, we extracted DNA from PBMCs that comprise a heterogeneous cell population. Since uneven cell type composition can lead to spurious results, we had to adjust our data for cell type composition.

We recognize that our cohort comprises different MS types and therefore our main analysis can identify DNA methylation changes that are present in most of the MS-affected MZ co-twins (i.e. RRMS or present in all MS types). In order to identify DNA methylation changes that are specific for SPMS or PPMS, a stratified analysis has to be carried out only including the twin pairs of which the MS-affected co-twin has SPMS or PPMS. However, the number of SPMS or PPMS cases in our cohort are too small to examine DNA methylation changes associated with these MS subtypes.

3. WP-DMR analysis is highly relevant and interesting, however, $\Delta\beta$ -value “> 0.20 and the aberrant methylated co-twin having a β -value greater than ± 3 ” might be too stringent given the heterogeneous cellular source (PBMCs) and the fact that finding differences typical for a subgroup of MS patients (not an individual pair) would be most relevant in terms of interpretability. The less stringent criteria (e.g. lower $\Delta\beta$) should be at least considered when looking for evidence of a WP-DMR in other twin pairs. However, it would be highly desirable to see this analysis where the focus is on finding within a subgroup differences (would likely require less stringent $\Delta\beta$).

Answer: PBMCs comprise a heterogeneous cell population and therefore we applied relatively stringent filtering criteria for the identification of the WP-DMRs. However, as suggested by the reviewer we applied a lower $\Delta\beta$ -value threshold of 0.15 in order to evaluate whether the 27 WP-DMRs, which were aberrantly methylated in the MS-affected co-twins, were present in other pairs as well. In total, four WP-DMRs were also identified in other twin pairs, of which one intergenic WP-DMR was present in 4 pairs and always hypermethylated in the MS affected co-twins (Supplementary Table 5). Hence, in total we identified 24 MS-associated WP-DMRs in 11 pairs, of which 23 were pair-specific and one present in 4 twin pairs We added this to the Results section on page 8 and to the Discussion at page 14.

4. Have authors attempted to validate FIRRE (which candidate CpG is not present on the EPIC array) in the PBMC cohort of 45 twins with TDBS (or other technique)?

Answer: Although the CpGs within the *FIRRE* MS-DMR are indeed not covered by the EPIC array, the cg08117231 probe, located only 6 bp upstream of this DMR, was not significant in the PBMC-based EWAS ($P > 0.05$). Accordingly, we do not expect that TDBS of the *FIRRE* MS-DMR in PBMCs will reveal a significant difference. This might indicate that the *FIRRE* MS-DMR is restricted to CD4+ T cells. Unfortunately, we are not able to test this hypothesis, because we do not have CD4+ T cells available of the other twins.

To be more precise we changed in the results at page 7 “located close to this DMR” into “located 6 bp upstream of this DMR”. We also added that we did not observe a significant difference, when we only included female samples in the PBMC-based EWAS.

In addition, we added the following sentences to the discussion (also as a response to question 5 of reviewer #3)(page 14): “Although the CpGs within this MS-DMR are not covered by the EPIC array, a probe located 6 bp upstream of the DMR did not show a significant difference in the PBMC-based EWAS. This might indicate that this MS-DMR is CD4+ T cell-specific, but it can also be the result of stochastic variation caused, for example, by molecular processes such as X-inactivation.”

5. Line 260: “slightly unbalanced, because 59.1% and 55.9% of the CpGs have”. What is this balance if $\Delta\beta$ cut-offs are introduced together with more stringent p-values. Does it still favour “predominant” hyper-methylation?

Answer: In Supplementary Table 9 we added the % of hypermethylated CpGs having $\Delta\beta > 0$, $\Delta\beta > 0.005$ and $\Delta\beta > 0.01$, with and without P -value < 0.001 in the analysis unadjusted and adjusted for cell-type composition. Introduction of these $\Delta\beta$ cut-offs and a P -value threshold did not change our results, because in all conditions the % of hypermethylated CpGs are higher in the MS-affected co-twins. We refer to this table in the main text (page 11).

6. Correlation presented in Suppl. Fig. 15B is not very convincing. More evidence should be provided or discussion regarding GC induced hypermethylation revisited.

Answer: We agree that the proposition of a link between glucocorticoid (GC) treatment and genome-wide hypermethylation requires more evidence, therefore, we critically reviewed our statements and rephrased our interpretations more carefully (page 15):

“While in our data, GC treatment history was not directly associated with global hypermethylation, we observed a rather weak, but significant association between increased within-pair *ZBTB16* methylation differences and the number of hypermethylated CpGs in the MS-affected co-twins. Although further confirmation is needed, this association might indirectly indicate that GCs also affect global DNA methylation levels. This might also explain the strong repetitive element hypermethylation in MS patients reported by Neven et al.⁶², who applied an inclusion criterion of only >1 month after GC treatment. In our study, on the other hand, global hypermethylation in the MS-affected co-twins was only observed in the EPIC array data, and repetitive elements are strongly underrepresented on this array. Accordingly, additional studies are warranted to assess the association between global hypermethylation and MS and whether this association is confounded by GC treatment history.”

7. While IFN-DMPs, although not significant, demonstrate a convincing IFN signature (i.e. high overlap with IFN genes and IFN-response related GO terms), the genes associated with GC-DMPs don’t seem to overlap with known GC-response signatures. Please comment. Is *ZBTB16* methylation reported in other MS studies? Do other studies report IFN-DMPs? One would expect some. Please comment.

Answer: Regarding the *ZBTB16* DMP we added the following to the discussion (page 12):

“The most prominent MS-DMP in our EWAS was the technically replicated cg25345365 DMP in *ZBTB16*, which has thus far not been reported in other MS EWAS studies.²⁵⁻³¹ This is, however, not surprising because the 450K array used in those studies does not cover this CpG.”

In addition, we added (page 12):

“Compared to the IFN analysis, we did not observe a very strong GC signature in the EPIC array data or in the WGBS data because in other common, GC-regulated genes, such as FKBP5, TSC22D3 and DUSP1^{55,59,60}, no DNA methylation differences were observed. This might be due to the fact that the IFN treatment was ongoing at the time of blood collection, while the GC treatment had been given >3-12 months prior to blood collection. On the other hand, a MS-affected co-twin very recently treated with GC was included in the WGBS analysis, but as this concerned a single replicate experiment, we had to apply very stringent analyses criteria (e.g., coverage threshold ≥ 15 reads).”

Concerning the IFN-DMPs, we are so far to our own surprise the first study that reports IFN-DMPs in MS. We added the following to the main text (page 9)

“In our cohort interferon-beta (IFN) is the most common disease-modifying treatment, and although IFN-induced transcriptomic alterations in blood and PBMCs of MS patients have been studied quite extensively⁵¹⁻⁵³, DNA methylation changes have not been reported so far.”

8. Line 35: “argues against some previously reported MS-associated epigenetic candidates”. This statement should be removed as it is imprecise (what previously reported MS-associated candidates have been excluded by the findings of this study?) and because the study design is considerably different from previous studies.

Answer: We agree with the reviewer that this is an imprecise statement and removed it from the abstract.

Minor issues:

1. Line 63: “Although the molecular mechanisms underlying these associations remain unknown, it is possible that these environmental factors operate by inducing DNA methylation changes.” Please indicate if DNA methylation changes in MS in relation to these factors have been reported.

Answer: For these three environmental risk factors (i.e., smoking, vitamin D and Epstein-Barr virus infection) several studies evaluating their effect on the DNA methylome have been published, of which some focused on MS. Hence, in the revised manuscript we re-formulated this sentence as follows and included appropriate references (page 3):

Although the molecular mechanisms underlying these associations remain unknown, evidence that these environmental factors can induce DNA methylation changes is accumulating.²¹⁻²⁴

For smoking we cite a meta-analysis that evaluates the effect of smoking on DNA methylation in blood using 450K data of 15907 individuals (Joehanes *et al.*, *Circ Cardiovasc Genet*, 2016), and we cite a study that examines the effect of smoking on DNA methylation levels in blood cells from MS patients (Marabita *et al.*, *Scientific Reports*, 2017). Concerning vitamin D, we cite a study of Zeitelhofer *et al.* (*Proc Natl Acad Sci USA*, 2017) who observed in mice with experimental autoimmune encephalomyelitis (a multiple sclerosis model) that vitamin D supplementation causes a genome-wide reduction of DNA methylation in CD4+ T cells. In addition, we cite a review of Scott *et al.* (*Curr Opin Virol*, 2017) that summarizes evidence on how epigenetic manipulation by the Epstein-Barr virus (EBV) can affect the host epigenome/methylome and causes EBV-associated cancers.

2. Line 111: Please clarify what is a suggestive P-value $<5*10^{-6}$ based on.

Answer: This is an arbitrary threshold and we added this to text in the Material and Methods (page 19). *“For the MS EWAS, an arbitrary significance level $\alpha < 5*10^{-6}$ was considered suggestive and genome-wide significance was defined as false discovery rate (FDR) < 0.05 .”*

3. Lines 133-135 “Next, a functional annotation analysis carried out with all MS-DMPs having a P <0.001 in the analysis adjusted for cell-type composition (698 in total) revealed TMEM232 as a prime candidate marker for MS. All other annotation categories were not significant.” Please clarify what do these analysis comprise and refer to. It is not clear.

Answer: This issue has also been raised by reviewer #3 and we edited this section as follows (page 6):

“Next, all MS-DMPs with a p -value < 0.001 in the analysis adjusted for cell-type composition (698 in total) were functionally annotated using the GREAT tool, which assigns biological meaning to a set of non-coding genomic regions by analyzing the annotations of the nearby genes⁴³. This analysis revealed that *TMEM232* is enriched for MS-DMPs. None of the other annotation categories were significant.” The settings of the GREAT tool are described in the Material and Methods (page 19)

4. Line 173: “45 WP-DMRs were identified in 17 out of the 45 twins”. Is there something in common for MS of these 17 twins i.e. MS type, disease duration etc.?

Answer: As suggested in Point 3 of the Major Issues, we evaluated whether the 27 WP-DMRs, that were aberrantly methylated in the MS-affected co-twins, were present in other pairs as well by applying a lower $\Delta\beta$ -value threshold of 0.15. In total, four WP-DMRs were also identified in other twin pairs, of which one intergenic WP-DMR was present in 4 pairs and always associated with the MS phenotype. Hence, in total we identified 24 MS-associated WP-DMRs in 11 pairs. We checked whether these 11 pairs differ in clinical characteristics such as, gender, MS course, disease duration at sampling date, age at first disease manifestation, MS treatment, and pack-years at sample collection, but we did not observe any significant differences. These results are summarized in Supplementary Table 5.

5. DMPs in *TMEM232* are possibly the only truly MS-associated changes. Can one infer something from regulatory region annotation in primary immune cells? Is there any overlap with other methylation studies in MS (even among suggestive loci with same directionality)?

Answer: We added Supplementary Figure 20 in which we show the status of the active H3K4me3 and repressive H3K27me3 chromatin marks at the *TMEM232* promoter region in different immune cell types from BLUEPRINT samples. We refer to this figure in the discussion (page 13) as follows:

*“The *TMEM232* MS-DMR is in close vicinity of the transcriptional start site and shows in different immune cell types enrichment for the chromatin activation mark H3K4me3 (Supplementary Fig. 20), hence, the observed methylation changes might be associated with expression changes,…”*

As requested by reviewer #1 as well, we evaluated the *TMEM232* MS-DMPs in whole blood-based 450K EWAS data of 140 unrelated MS patients and 139 controls from Kular et al. (*Nature Communications*, 2018). Unfortunately, the cg27037608 and cg26583412 *TMEM232* MS-DMPs were not present on the 450K array, but data from seven neighboring CpGs were available. Although none of these CpGs was significantly differentially methylated between the MS cases and controls ($P > 0.05$), the methylation levels were always higher in the MS patients, confirming the directionality of the effect observed in the twin cohort (see Supplementary Table 2). We added this to the Results section on page 7. In addition, we added the following to the discussion on page 13:

*“In whole blood-based case-control 450K data no significant difference was observed, but the two most prominent *TMEM232* MS-DMPs were not present on the 450K array. Nevertheless, all *TMEM232* CpGs present on the 450K array confirmed the directionality of the effect observed in the twins, which might indicate that the MS-DMR is restricted to a PBMC subtype and is diluted in whole blood in which neutrophils are the predominant cell type.”*

6. What is the regulatory region annotation/chromatin state of *FIRRE* in various primary CD4 subsets (Blueprint data)?

Answer: We added Supplementary Figure 10 in which we show the methylation and chromatin status of the *FIRRE* MS-DMR in various subsets of primary CD4+ T cells in male and female BLUEPRINT samples. The BLUEPRINT whole genome bisulfite sequencing data of central memory and effector memory CD4+ T cells shows that in females methylation levels at the *FIRRE* DMR locus are lower compared to males (~50% versus ~100%). In addition, in female central memory CD4+ $\alpha\beta$ T cells a H3K4me3 peak is observed at the *FIRRE* DMR locus, but this peak is absent in the male sample. We refer to this figure in the results section (page 7) and in the discussion (page 14).

7. WGBS in CD4+ cells: what is the rationale for using a mean difference of 20%, which is rather large, especially for analysis in a sorted cell type? Is there more overlap with EPIC data or other published data in CD4 with less robust cut-off for a difference?

Answer: For the WGBS in CD4+ T cells we had 4 female twin pairs available. Due to this small sample size, the quality of the DMR analysis on the WGBS data is highly dependent on the sequencing coverage. Since we applied the general used coverage threshold of ≥ 10 reads in all samples, applying a mean difference threshold of only 10 to 15 % would result in too many false positives. Accordingly, we applied a mean difference threshold of 20%. Nevertheless, as requested we now also performed a DMR analysis using a less robust cut-off difference of 15%, which resulted in 19 additional MS-DMRs. Although five of these MS-DMRs were also covered by the EPIC array, they were not significant in the PBMC-based EWAS data (see Supplementary Table 3). Remarkably, another MS-DMR was located in the *DDAH1* gene that also contains an established MS-associated SNP (Sawcer *et al.*, *Lancet Neurol*, 2014). We describe the results of this additional MS-DMR analysis at page 7.

8. Line 357: “Consequently, our data does not support a contribution of genomic imprinting errors in the pathogenesis of MS.” This is too strong statement considering that specific cell types, additional modifications and mechanisms of imprinting have not been investigated.

Answer: The reviewer is correct and we adapted this sentence as follows (page 15):
“Consequently, our PBMC-based data do not support the hypothesis that genomic imprinting errors contribute to the discordant clinical manifestation of MS in these MZ twins.”

Reviewer #3:

Souren *et al.* have investigated Multiple Sclerosis in Monozygotic (MZ) disease-discordant twin pairs by analysis of the DNA methylation with the Illumina EPIC (850k) arrays in 45 pairs and Whole Genome Bisulphite Sequencing (WGBS) in 4 pairs. The former was in peripheral blood mononuclear cell-derived DNA and the later in isolated CD4+ T cells.

On the whole the study is well designed and analysis is robust – with, as expected in an MZ discordant study, the removal of genetic confounding leading cell-type heterogeneity from blood-derived DNA to be the most considerable factor influencing variability. I have some queries below, which the authors could address, that I think would help improve and qualify the results as presented. In particular, points regarding the targeted sequencing to validate the findings and the assessment of the DNA methylation state of repetitive elements.

Major

1) The references given for the identification of discordant DNA methylation profiles for MZ twins in imprinted loci are at least a decade old (pg 3, line 57) – are there any more recent and robust examples of this instead?

Answer: Discordant methylation profiles at imprinted loci in MZ twins have mainly been reported for the KvDMR (i.e., KCNQ1OT1 ICR), causing the Beckwith-Wiedemann syndrome (BWS) in the hypomethylated MZ co-twin. The majority of these publications are case reports, and only Weksberg *et al.* (2002) and Blik *et al.* (2009) described a larger subset of cases (10 MZ twin pairs each). Since *Nature Communications* allows maximum 70 references, we only referred to Weksberg *et al.* (2002), but now we realize that this gives an outdated impression. In the revised manuscript we refer to the “large” study of Blik *et al.* (2009) and to a recent case report of Inoue *et al.* (2017) that also summarizes all studies on BWS discordant MZ twins including molecular data. Furthermore, we refer to a recent study of Riess *et al.* (2016) that describes two MZ twin pairs of which one is concordant and the other is discordant for Silver-Russel syndrome (SRS) due to hypomethylation at the H19/IGF2 ICR. In addition, Riess *et al.* (2016) summarizes all the previously published cases of MZ twins discordant for SRS.

2) The authors should include more discussion regarding the age distribution of the twins and years clinically discordant between the pairs. Differences in the disease age-of-onset can be a significant issue in MZ discordant studies. The range of 0 – 45 years for clinical discordance in this study (Table 1) indicates some twins have only recently been diagnosed. Therefore, there is the increased potential for some of these MZ pairs to become concordant over time. The authors should explore how this may have influenced their analysis.

Answer: As requested by the other reviewers as well, we performed a stratified analysis only including the 25 pairs that have been clinically discordant for MS for more than 10 years (see Supplementary Table 1). In the analysis adjusted for cell-type composition, two DMPs had a suggestive P-value $< 5 \times 10^{-6}$, one of which is located in the promoter region of the *TACSTD2* gene encoding tumor-associated calcium signal transducer 2 (mean $\Delta\beta$ -value = -0.022). The second DMP is located in the promoter region of the *RCL1* gene (mean $\Delta\beta$ -value = -0.011), which has been linked to depression. In addition, the *TMEM232* MS-DMPs cg27037608 (mean $\Delta\beta$ -value = 0.026, $P = 3.2 \times 10^{-5}$) and cg26583412 (mean $\Delta\beta$ -value = 0.038, $P = 1.8 \times 10^{-5}$) were among the top 15 most significant DMPs associated with long-standing MS. Furthermore, the *ZBTB16* MS-DMP had a mean $\Delta\beta$ -value difference of -0.036 and $P = 0.002$ (Supplementary Table 1). These results indicate that the *TMEM232* and *ZBTB16* MS-DMPs are also associated with long-standing MS. We added this to the Results section on page 6 and mention it in the Discussion at page 13.

In addition, as a response to question 1 of reviewer #2 as well, we comment on the issue that some of the pairs included in our study might get clinically concordant for MS in the future in the Discussion at page 16.

3) The study of Baranzini *et al.* is mentioned in the Introduction (pg 2, line 79) - it should also be included in the text that this study's analysis threshold required a near total reversal of methylation state to identify significant CpGs – that is a change between discordant siblings from $\leq 20\%$ to $\geq 80\%$ methylated, or the reverse.

Answer: We adapted this section in the introduction as follows (page 4):

“Thus far, one EWAS comprising three MS discordant MZ twins has been reported in which no DNA methylation differences were identified³⁴. Due to the small sample size, that study could only detect very large DNA methylation differences, and therefore further studies in larger cohorts are required”.

4) The “functional annotation analysis” (pg 5, line 133) should be described in more detail here – no reference to method given or description?

Answer: The same issue has also been raised by reviewer #2 and we edited this section as follows (page 6):

“Next, all MS-DMPs with p -value <0.001 in the analysis adjusted for cell-type composition (698 in total) were functionally annotated using the GREAT tool, that assigns biological meaning to a set of non-coding genomic regions by analyzing the annotations of the nearby genes⁴³. This analysis revealed that TMEM232 is enriched for MS-DMPs. All other annotation categories were not significant.”

The settings of the GREAT tool are described in the Material and Methods (page 19)

5) Could the authors comment on whether stochastic variation in DNA sample and random X inactivation could influence the result found within the *FIRRE* gene?

Answer: We added the following sentences to the discussion (also as a response to question 4 of reviewer #2)(page 14):*“...Although the CpGs within this MS-DMR are not covered by the EPIC array, a probe located 6 bp upstream of the DMR did not show a significant difference in the PBMC-based EWAS. This might indicate that this MS-DMR is CD4+ T-cell-specific, but it can also be the result of stochastic variation caused, for example, by molecular processes such as X-inactivation.”*

6) The authors state that DNA methylation changes were found in the imprinted genes *SVOPL* and *HMI3* (pg 7 line 183). Was this within the Imprinting Control Region (ICR) of either gene?

Answer: The *HMI3/MCTS2P* within-pair differentially methylated region (WP-DMR) (chr20:30,134,929-30,135,362 (hg19)) is within the imprinted DMR that is located in the promoter CpG island of the maternally imprinted and paternally expressed *MCTS2P* gene (also known as *PSIMCT-1*). The *SVOPL* WP-DMR (chr7:138,348,774-138,349,443 (hg19)) overlaps with the imprinted DMR situated in a CpG island in the promoter region of the short isoform of *SVOPL*. Thus far, it has not been reported whether these imprinted DMRs also function as imprinted control regions, since this has to be confirmed by disruption of the imprinted expression upon genetic deletion of that DMR (either through experimental targeting in mouse or by a spontaneously occurring deletion in humans (Court *et al.*, Genome Research, 2014)). Nevertheless, to be more precise in the text, we changed “*imprinted regions*” into “*imprinted DMRs*” (page 8 and 14).

7) Cloning was employed to reduce PCR bias with traditional Sanger Bisulphite sequencing. For the same reason, Targeted Bisulphite sequencing methods have used micro-fluidic or droplet methods to minimise potential bias, with, for example, Raindance¹ or Fluidigm² methodology. The authors should comment on the drawbacks of not using such an approach here. Whilst results appear consistent between array and targeted analysis, the $r=0.8$ – so therefore, not exact, which is potentially indicated by the larger methylation range for the TDBS in Figure S4 *TMEM232* cg27037608 for example. This imprecision in the method may be more biased in some CpG locations.

Answer: To minimise PCR-bias in our targeted deep bisulfite sequencing (TDBS) analysis we applied: a high quality hot start DNA polymerase (Qiagen), an adequate amount of bisulfite-treated input DNA (40 ng in 30 μ l reaction), low primer concentrations to reduce primer dimer formation (133 nM) and a

high annealing temperature to increase specificity (60°C for the *TMEM232* and *ZBTB16* amplicons). In addition, the amplicons were deeply sequenced with a minimum coverage of 1500 reads. Hence, our TDBS data correlates very well with the EPIC array data, with $r=0.84$ and $r=0.89$ for *TMEM232* and *ZBTB16*, respectively. But most importantly, the TDBS data confirms that these DMPs are significantly differentially methylated between the clinically non-affected and the MS-affected co-twins, even though the mean methylation differences observed for the *TMEM232* and *ZBTB16* DMP are very small (2% and 4%, respectively) and therefore hard to replicate. For *TMEM232*, we now also included the TDBS data of the other 8 CpGs that were present in the *TMEM232* amplicon, which shows that the neighboring CpGs are also significantly differentially methylated (Supplementary Figure 5). Accordingly, we do not share the opinion that our TDBS data is imprecise.

However, we would like to note that when two methods are compared a correlation coefficient can be a misleading measure, because it is highly dependent on the total variation. This means that if the selected locus would be highly variably methylated across the samples and covers a wide methylation range (0 - 1), then a very high correlation coefficient can be observed. However, in our case the *TMEM232* and *ZBTB16* DMPs do not cover a very wide methylation range (0.31-0.59 and 0.36-0.72, respectively). In addition, although the EPIC array generates high quality methylation data, it also has limitations and cannot be considered as the gold standard. For example, the protocol contains a whole genome amplification step, and the EPIC array β -value is the ratio of the methylated probe intensity and the overall intensity, and is thus only a methylation index. Therefore, the larger methylation range observed in the *TMEM232* TDBS data, can also indicate that the EPIC array data is a bit biased.

8) The use of generic primers to assay the repetitive regions, Alu, HERVK, and LINE1, which are exceeding numerous throughout the genome could lead to stochastic findings unless extreme depth of coverage was attained. There are, for example, >1 million copies of Alu elements throughout the genome. Can the authors comment on the appropriateness of this generic primer methodology and whether the results were reproducible? Additionally, do the primers preferentially assay a subfamily set of any of the repeat families, due to any sequence variation within the primers?

Answer: The *HERVK* primers were specifically designed to amplify LTR5Hs, which is the youngest subfamily. *In silico* bisulphite PCR using BiSearch (<http://bisearch.enzim.hu>, not allowing mismatches) demonstrates that our *HERVK* primers amplify 328 different PCR products, of which 98% matches to LTR5Hs. *LINE1* primers were designed using the promoter/5'-UTR consensus sequence (GenBank-Nr. X58075.1). *In silico* these *LINE1* primers generate 309 different specific PCR products, which also mainly comprise the youngest subfamilies L1HS (~64%), L1PA2 (~25%) and L1PA3 (~9%). The Alu primers were designed using the consensus sequence according to Price *et al.* (Genome Research, 2004) and generate *in silico* an “infinite” number of different PCR products.

The repetitive elements were sequenced with a minimum coverage of 2000 reads, giving a >6 fold coverage per individual *HERVK* and *LINE1* element. However, as remarked by the reviewer, this coverage is only a “snapshot” for the much greater number of Alu elements. Nevertheless, the results were reproducible, which is in our twin design translated in small maximum (absolute) within-pair methylation differences of only 0.015, 0.024 and 0.025 for Alu, *HERVK*, and *LINE1*, respectively.

All details on the primer specificity were added to the legend of Supplementary Table 11, in which the primer sequences and PCR conditions of the repetitive elements are listed. Details on the reproducibility of the assays were added to the results section (page 11) .

9) Whilst the value of assessment of global DNA methylation is debated beyond examples of gross cancer tissue differences, the authors should acknowledge they are usually performed on total genomic DNA. Therefore, are strongly driven by the methylation state of the repetitive portion of the genome. The EPIC array still only assays ~3% of the CpGs within the DNA methylome and is bias towards the more unmethylated and less repetitive portion than the true genome-wide picture. Consequently, the authors should acknowledge this shortcoming in any inference of the global DNA methylation state in the disease-related twin.

Answer: We agree with the reviewer and we acknowledge this in the Discussion section as follows (page 15):

“In our study, on the other hand, global hypermethylation in the MS-affected co-twins was only observed in the EPIC array data, and repetitive elements are strongly underrepresented on this array. Accordingly, additional studies are warranted to assess the association between global hypermethylation and MS and whether this association is confounded by GC treatment history.”

10) The Discussion correctly states that population studies have considerable potential for genetic confounding so that genetic effects will be driving many of these observations – see also recent papers^{3,4} on the extent of this potential genetic confounding, both within array and in sequencing based-studies, respectively. This point regarding genetic confounding in population-based studies should be strongly reiterated.

Answer: We now moved this part towards the end of the discussion at page 15 and added the findings of Hannon et al. (*PLoS Genetics*, 2018). The section reads now as follows:

“None of the MS-DMPs observed in our study have previously been reported by other EWAS for MS²⁵⁻³¹. However, those MS EWAS studies observed much larger methylation differences and used selection criteria for MS-DMPs of absolute mean β -value differences $>0.05^{25}$, or even $>0.10^{26-29}$. As these other studies used genetically unmatched cases and controls²⁵⁻³¹, the large methylation differences they observed might mainly be driven by genetic variation. This is supported by a recent study of Hannon et al.⁶⁸, who showed that, in particular, those sites with variable DNA methylation levels and those that have robustly been associated with environmental exposures are influenced by genetic effects, highlighting the need to control for genetic background in EWAS.”

11) The observation that the >3 month cut off for GC treatment (pg 10 line 293) was not sufficient is of significant interest. If this could be further supported/replicated the authors could consider including this in the Abstract.

Answer: We agree with the reviewer that this an important finding. Since none of the MS-affected co-twins included in the array-based EWAS received glucocorticoids (GCs) within three months prior blood collection, our data shows that a >3 month cut off is not sufficient. Unfortunately, we are now not able to further replicate these results in other cohorts, because the *ZBTB16* DMP is not present on the 450K array and at the moment there is no EPIC array data of MS patients and controls publicly available. Nevertheless, for a proper validation a longitudinal study design is required, that (ideally) includes GC-treatment naive MS patients providing blood samples before treatment and then serial samples at monthly intervals during the first year period and quarterly intervals during the second year after treatment.

Minor

1) The inclusion of “Large cohort” in the title - The authors should either state the precise number of twin pairs or remove. Size comparators date very quickly, and meta-analyses of discordant twin sets are or will be performed very shortly⁵, even for comparatively rare diseases, as in this case.

Answer: We removed “a large cohort” from the title.

2) The Abstract is too vague in a number of places and should be re-written to be precise in its details *i.e.* – ‘a few MS-associated DMPs’, ‘many regions’ – instead state exact numbers.

Answer: We added the exact numbers to the abstract.

1. Paul, D.S. *et al.* Assessment of RainDrop BS-seq as a method for large-scale, targeted bisulfite sequencing. *Epigenetics* 9(2014).
2. Korbic, D. *et al.* Multiplex bisulfite PCR resequencing of clinical FFPE DNA. *Clinical Epigenetics* 7, 28 (2015).
3. Hannon, E. *et al.* Characterizing genetic and environmental influences on variable DNA methylation using monozygotic and dizygotic twins. *PLoS Genet* 14, e1007544 (2018).

4. Bell, C.G. *et al.* Obligatory and facilitative allelic variation in the DNA methylome within common disease-associated loci. *Nat Commun* 9, 8 (2018).
5. Willemsen, G. *et al.* The Concordance and Heritability of Type 2 Diabetes in 34,166 Twin Pairs From International Twin Registers: The Discordant Twin (DISCOTWIN) Consortium. *Twin Res Hum Genet* 18, 762-71 (2015).

Reviewer #1 (Remarks to the Author):

The majority of concerns have been addressed, however I have two further comments in response to the authors' reviewer responses to points 2 and 7 raised in the initial review.

2) The authors mention how Paul et al performed DVP analysis in individual cell types, which is indeed a very valid approach with increased power to detect variability compared to heterogeneous tissues such as PBMCs. However, recently a study which used whole blood samples of rheumatoid arthritis discordant monozygotic twins identified a DVP signature which provided insights into disease aetiology [1], despite the use of heterogeneous tissue. Thus, it could still be of interest to investigate differentially variable positions in this cohort.

In line 477 of revised manuscript – 'differentially variable methylated regions' should be changed to 'differentially methylated regions' as this is what the analysis describes, no variability analysis has been performed.

7) The reviewer recognises that investigation of the effect of smoking on DNA methylation in general is not within the scope of the study, however this is not what was requested. Despite being balanced between comparison groups, smoking status could indeed confound the described pairwise analyses due to its strong influence on DNA methylation. Could the authors clarify if smoking status was correlated with DNA methylation measurements during principal components analysis, and if adjustment for smoking status alters the outcome of the DMP/DMR analyses?

1. Webster AP, Plant D, Ecker S, Zufferey F, Bell JT, Feber A, Paul DS, Beck S, Barton A, Williams FMK, Worthington J: Increased DNA methylation variability in rheumatoid arthritis-discordant monozygotic twins. *Genome Med* 2018, 10:64.

Reviewer #2 (Remarks to the Author):

The authors have addressed the most important concerns in a satisfactory manner.

Reviewer #3 (Remarks to the Author):

Souren *et al.* have replied on the whole to my comments successfully. However, there remains a few points that require further clarification, detailed below.

Major

3. The authors have misinterpreted my comment – please refer back to the paper of Baranzini *et al.* Those authors chose those levels as their analytic cut-offs and did not report or attempt to identify any changes less than this size even if they existed in their dataset (*e.g.* would not have identified complete loss of imprinting.)

8. The authors should include in the manuscript the caveats about these generic primer repeat results, particularly regarding the very small and stochastic interrogation of the large Alu-repeat space of the genome.

9. The authors should not refer to ‘global’ methylation when assessed by array – for the reasons already detailed – instead could refer to this as ‘total EPIC array’ methylation.

Reviewers' comments

Reviewer #1:

The majority of concerns have been addressed, however I have two further comments in response to the authors' reviewer responses to points 2 and 7 raised in the initial review.

Answer: We are pleased that we sufficiently addressed the majority of the raised concerns. Please find below our answers to the remaining issues.

2) The authors mention how Paul et al. performed DVP analysis in individual cell types, which is indeed a very valid approach with increased power to detect variability compared to heterogeneous tissues such as PBMCs. However, recently a study which used whole blood samples of rheumatoid arthritis discordant monozygotic twins identified a DVP signature which provided insights into disease aetiology [1], despite the use of heterogeneous tissue. Thus, it could still be of interest to investigate differentially variable positions in this cohort.

Answer: The reviewer is correct and as suggested we tested in our twin cohort whether DNA methylation variability is also implicated in MS. For this DVP analysis, probes containing a SNP within five bases of the measured CpG site, probes mapping to the sex chromosomes, and probes with at least one missing value were excluded, which resulted in methylation data of 759,291 sites. DVPs were identified using the iEVORA algorithm, which measures differential variability between two groups by utilizing the Bartlett's test to detect differences in variance and an unpaired T-test to identify difference in means. In agreement with Paul et al. (Nature Communications, 2016) and Webster et al. (Genome Medicine, 2018), DVPs were defined as CpGs with a FDR-corrected Bartlett's P-value < 0.001 and raw T-test P-value < 0.05. However, this approach revealed only 25 differentially variable CpG positions (DVPs) of which the majority (88%) was hypervariable in the non-affected MZ co-twins (see Supplementary Table 6 and Supplementary Figure 15), which argues against increased DNA methylation variability in MS-discordant MZ twins.

The methods used for the DVP analysis are described at page 20 of the revised manuscript, and we added the following paragraph entitled “No increased DNA methylation variability in MS-discordant twins” to the results section at page 9: “Increased DNA methylation variability has been observed in MZ twins discordant for the autoimmune diseases type 1 diabetes (T1D) and rheumatoid arthritis (RA).^{12,13} Hence, we tested whether DNA methylation variability is also implicated in MS using the iEVORA algorithm.⁵² However, applying the default FDR < 0.001 resulted in only 25 differentially variable CpG positions (DVPs) of which the majority (88%) was hypervariable in the non-affected MZ co-twins (see Supplementary Table 6 and Supplementary Figure 15). Hence, our PBMC-based EPIC array data does not support the presence of an MS-associated DNA methylation variability signature in these MS-discordant MZ twins.”

In addition, we added the following to the discussion at page 17.

“...Paul et al.¹² identified 10,548 differentially variable CpG positions (DVPs) in B cells, 4,314 in CD4+ T cells, and 6,508 in monocytes, and the T1D-affected MZ co-twins were enriched for DVPs. In addition, Webster et al.¹³ identified 1,107 DVPs in whole-blood 450K data of 79 RA-discordant MZ twins, of which 763 DVPs were hypervariable in the RA-affected MZ co-twins. Although, we used the same method and significance threshold as applied in these studies, we only identified 25 DVPs of which the majority was hypervariable in the non-affected co-twins. Accordingly, our PBMC-based EPIC data does not reveal an MS-associated DNA methylation variability signature.”

In line 477 of revised manuscript – ‘differentially variable methylated regions’ should be changed to ‘differentially methylated regions’ as this is what the analysis describes, no variability analysis has been performed.

Answer: This section has now been rewritten to discuss the results of our DVP analysis (see above), and the term ‘differentially variable methylated regions’ is removed.

7) The reviewer recognizes that investigation of the effect of smoking on DNA methylation in general is not within the scope of the study, however this is not what was requested. Despite being balanced between comparison groups, smoking status could indeed confound the described pairwise analyses due to its strong influence on DNA methylation. Could the authors clarify if smoking status was correlated with DNA methylation measurements during principal components analysis, and if adjustment for smoking status alters the outcome of the DMP/DMR analyses?

Answer: To address the concern of the reviewer, we repeated the pair-wise association analysis on the EPIC array data of the 45 MZ twin pairs by including smoking status (at sample collection) as an additional covariate in the adjusted analysis. The P-values of our 5 top MS-associated DMPs listed in Table 2 (including the *ZBTB16* and *TMEM232* MS-DMPs) were $<7*10^{-6}$, indicating that our results are not confounded by smoking status. Hence, we added the following sentence to the Results section at page 6: “When including smoking status as an additional covariate in the adjusted analyses, P-values of these 5 MS-DMPs were $<7*10^{-6}$ indicating that our results are not confounded by smoking status”.

We also repeated the pair-wise analysis in which we only included the EPIC array data of the 25 pairs that have been clinically discordant for MS for more than 10 years (long-standing MS), and included smoking status (at sample collection) as an additional covariate in the adjusted analysis. However, additional adjusting for smoking status barely affected the results of the association analysis and the top 15 “long-standing MS-DMPs” did hardly change. The two *TMEM232* MS-DMPs and the *ZBTB16* MS-DMP became even slightly more significant. Hence, we included the following sentence at the bottom of page 6 of the Results: “Additional adjustment for smoking status did not change the results”.

In our study, we also performed an MS-DMR analysis on WGBS data of CD4+ memory T cells of four MS-discordant female MZ twin pairs, in which we identified an MS-DMR in the *FIRRE* gene. This MS-DMR analysis cannot be adjusted for smoking status, because with a sample size of eight samples in total it is not possible to properly estimate the effect of smoking on DNA methylation levels. Since the number of smokers and non-smokers at sample collection are perfectly balanced between the MS-affected (n=2) and non-affected MZ co-twins (n=2) included in the WGBS analysis (see also Supplementary Table 13), we consider it unlikely that the MS-DMR analysis is confounded by smoking status. Nevertheless, we checked whether smoking-related methylation changes have been observed at the *FIRRE* locus by examining the results of the meta-analysis of Joehanes et al. (*Circ Cardiovasc Genet*, 2016), who evaluated the effect of smoking on DNA methylation in blood using 450K data of 15,907 individuals. Comparing current versus never smokers, Joehanes et al. observed 18,760 CpGs that were statistically significantly differentially methylated at FDR<0.05. However, the *FIRRE* locus was not among these 18,760 CpGs, which supports that the association between *FIRRE* methylation levels and the MS phenotype is not confounded by smoking status. For a matter of fact, none of the top 5 MS-DMPs listed in Table 2 (nor neighbouring CpGs) were among these 18,760 smoking-associated CpGs.

1. Webster AP, Plant D, Ecker S, Zufferey F, Bell JT, Feber A, Paul DS, Beck S, Barton A, Williams FMK, Worthington J: Increased DNA methylation variability in rheumatoid arthritis-discordant monozygotic twins. *Genome Med* 2018, 10:64.

Reviewer #2:

The authors have addressed the most important concerns in a satisfactory manner.

Answer: We are happy to hear that we adequately addressed all important concerns.

Reviewer #3:

Souren et al. have replied on the whole to my comments successfully. However, there remain a few points that require further clarification, detailed below.

Answer: We thank the reviewer for the positive evaluation. Please find below our response to the remaining concerns.

Major:

3) The authors have misinterpreted my comment – please refer back to the paper of Baranzini et al. Those authors chose those levels as their analytic cut - offs and did not report or attempt to identify any changes less than this size even if they existed in their dataset (e.g. would not have identified complete loss of imprinting).

Answer: The reviewer is correct and we now formulated this section in the introduction as follows (page 4): *“Thus far, one EWAS in MS-discordant MZ twins has been reported in which no DNA methylation differences were identified³⁵. Since this study included only three twin pairs and exclusively aimed at identifying very large methylation differences (i.e., $\geq 80\%$ methylation in one co-twin and $\leq 20\%$ in the other), further studies in larger cohorts are required.”*

8) The authors should include in the manuscript the caveats about these generic primer repeat results, particularly regarding the very small and stochastic interrogation of the large Alu repeat space of the genome.

Answer: To emphasize the limitations of the generic primer approach we added the following to the results section at page 11: *“Please note that for Alu generic primers were used, and since the element has over 1 million copies in the genome, with a minimum sequencing coverage of 2000 reads probably less than 1% of the elements were analysed. In contrast, HERVK and LINE1 primers were designed to amplify the youngest subfamilies, which gives with a minimum sequencing coverage of 2000 reads a >6 fold coverage per individual HERVK and LINE1 element (see the legend of Supplementary Table 12 for details).”*

9) The authors should not refer to ‘global’ methylation when assessed by array – for the reasons already detailed – instead could refer to this as ‘total EPIC array’ methylation.

Answer: The reviewer is correct and therefore we decided to avoid using the term “global” methylation. Instead we use “EPIC array-wide” methylation, when referring to the array data, and “repetitive element” methylation, when referring to the repeat data. Please see the changes marked in yellow at page 11-12 in the paragraph which is now entitled *“ZBTB16 methylation correlates with EPIC array-wide hypermethylation”*, and in the discussion at page 15.

Reviewer #1 (Remarks to the Author):

The authors have addressed the concerns raised in the previous review satisfactorily.

Reviewer #3 (Remarks to the Author):

The authors have responded to my points and included appropriate comments in the manuscript to address these clearly - thank you

REVIEWERS' COMMENTS:

Reviewer #1 (Remarks to the Author):

The authors have addressed the concerns raised in the previous review satisfactorily.

Reviewer #3 (Remarks to the Author):

The authors have responded to my points and included appropriate comments in the manuscript to address these clearly - thank you

Answer: We are happy to hear that we adequately addressed all concerns. We thank all reviewers for the very detailed evaluation of our manuscript, and the constructive comments and suggestions. The points raised have been very helpful for improving the manuscript.